

# Changes in water quality and ecosystem processes at extreme summer low flow of 2018 with high-frequency sensors

Jingshui Huang[1,2], Dietrich Borchardt[1], and Michael Rode[1,3]

[1]Department of Aquatic Ecosystem Analysis, Helmholtz Centre for Environmental Research – UFZ, Brueckstrasse 3a, 39114 Magdeburg, Germany

[2]Chair of Hydrology and River Basin Management, Technical University of Munich, Arcisstrasse 21, 80333 Munich, Germany

[3]Institute of Environmental Science and Geography, University of Potsdam, Karl-Liebknecht-Strasse 24–25, 14476 Potsdam-Golm, Germany

*Correspondence to*: Jingshui Huang (jingshui.huang@tum.de)

**Abstract.** The frequency and severity of summer droughts in Central Europe are expected to increase due to climate change, resulting in more frequent extreme summer low-flow events that significantly impact water quality and ecosystem processes. Despite the urgency of this issue, studies utilizing high-frequency measurements to analyze these effects remain scarce. This study focuses on the Lower Bode, a 27.4-km 6th-order agricultural stream in Saxony-Anhalt, Germany, equipped with 15-minute interval water quality measurement stations at both ends. The stream experienced extreme low-flow conditions during the summer of 2018. We compared water quality and ecosystem variables from 2018 to those of the 2014–2017 summers using the Kruskal-Wallis test. Results showed that water temperature and chlorophyll-a concentrations were significantly higher during the extreme low-flow event, while dissolved oxygen and nitrate concentrations were significantly lower. Diurnal dissolved oxygen fluctuations were more pronounced, with gross primary productivity (GPP) significantly elevated. Benthic algae were the dominant contributors to the increase in GPP (95%), with phytoplankton accounting for the remaining 5%. Ecosystem respiration also increased significantly, resulting in near-zero net productivity and a shift towards a less heterotrophic state. While net nitrate uptake rates remained consistent with previous years, the percentage of nitrate removed increased significantly, suggesting enhanced nitrate removal efficiency. This was driven by an increase in gross nitrate uptake, predominantly through benthic algae assimilation, highlighting a strengthened internal nutrient cycle during extreme low flow. Our findings provide new insights into water quality and instream ecosystem processes under extreme low-flow conditions, enhancing our understanding of potential future impacts under climate change.

## 1 Introduction

The 2018 drought in Germany and Central Europe set historic records (Mühr et al., 2018; Toreti et al., 2019) and has since become a reference event due to its widespread water deficits, ecosystem damage and significant crop yield loss (Mastrotheodoros et al., 2020; Toreti et al., 2019). Drought events are a key driver to low-flow conditions in rivers and streams (Van Loon, 2015). During the 2018 drought, water levels at many gauging stations in Germany reached historically low levels (BfG, 2019). Climate projections under high-emissions scenario suggest that





droughts with the intensity of 2018 could become a common occurrence in Central Europe as early as 2043 (Toreti et al., 2019). Furthermore, these projections predict an increase in the frequency and duration of summer low-flow periods in the region (Ionita and Nagavciuc, 2020).

While the impacts of extreme low flows on water quantity are well-documented, there are still knowledge gaps
regarding the effects on water quality and ecosystem processes in streams/rivers (Michalak, 2016; Mosley, 2015; Peña-Guerrero et al., 2020). Extreme low flows can affect the water quality by increasing residence times, decreasing water volumes/depths, reducing dilution of point-source pollution, and disrupting sediment, organic matter, and nutrient transport (Hensley et al., 2019; Mosley, 2015 and references therein). Additionally, these conditions are often accompanied by changes in critical environmental conditions of stream ecosystems, such as
rising temperature and increased solar radiation, which further intensifying their effects on instream water quality processes.

Most studies on water quality under extreme low-flow conditions rely on historical monitoring data (Hübner and Schwandt, 2018; Mosley et al., 2012; van Vliet and Zwolsman, 2008; Wilbers et al., 2009). However, such data are typically sparse and discrete—collected monthly, biweekly, or, at best, daily— due to the reliance on traditional
grab sampling methods by monitoring authorities. This limits the ability to capture short-term variability and understand the dynamics of extreme low-flow events (Mosley, 2015; Peña-Guerrero et al., 2020). The advent of high-frequency water quality monitoring using advanced sensor technology addressed this limitation, allowing for continuous data collection at fine temporal resolutions (e.g., 15-minute intervals). High-frequency monitoring reveals diel patterns, such as daily fluctuations in dissolved oxygen, temperature, and nutrient concentrations, driven
by biological activities like photosynthesis and respiration. These patterns are often smoothed out or entirely missed in low-frequency sampling (e.g., weekly or monthly), potentially leading to underestimations of extreme conditions or misinterpretation of system behavior. This fine-scale temporal resolution is crucial for understanding how ecosystems respond to transient events, such as extreme low flows, where rapid changes can significantly influence water quality and ecosystem processes. Gathering high-frequency data allows for better characterization of temporal
variability, decoupling short-term and long-term changes in water quality, providing insights into processes, and improving scientific understanding of contemporary environmental phenomena (Hamilton et al., 2015; Minaudo et al., 2018; Tran et al., 2022). To date, only few studies have explicitly addressed the water quality impacts of extreme summer low flows in European streams/rivers, based on high-frequency water quality monitoring campaigns despite the high scientific and social concerns due to the recent consecutive drought years (Addy et al., 2018; Hensley et al.,
2019; Hosen et al., 2019).

Previous research on drought and extreme low-flow impacts on water quality in streams/rivers has primarily focused on physical, chemical and biological indicators, such as water temperature, salinity, major ions, turbidity, pH, dissolved oxygen, nutrients, and algae (Mosley, 2015). However, the effects of extreme low flows on key instream ecosystem-level processes including ecosystem metabolism and nutrient uptake are still largely unknown. Evidence
suggests that during drought conditions, ecosystem metabolism, including gross primary production (GPP) and ecosystem respiration (ER), decreased in headwater streams but increased in large rivers and downstream estuaries (Addy et al., 2018; Bruesewitz et al., 2013; Crawford et al., 2017). Yet, the responses of high-order streams with a





transitional size, where phytoplankton and benthic algae might co-exist are less understood (Vannote et al., 1980). Furthermore, the individual contributions of phytoplankton and benthic algae to overall ecosystem metabolism and

their responses to extreme low-flow conditions have not been well-documented.

Nitrogen cycling is closely linked to ecosystem metabolism, particularly through processes like nitrogen assimilation and removal (Hensley et al., 2019). As GPP and ER drive the uptake and transformation of nitrogen in aquatic systems, changes in these metabolic rates can directly affect the efficiency of nutrient removal and internal recycling. For instance, increased GPP can enhance nitrate assimilation by autotrophs such as phytoplankton and

benthic algae, while ER contributes to nitrogen processing through heterotrophic pathways. However, the effects of extreme low flows on instream nutrient uptake, specifically nitrogen assimilation, nitrogen removal, and internal recycling, are poorly understood (Addy et al., 2018; Riis et al., 2017). Understanding these relationships is particularly important under extreme low-flow conditions, where nutrient dynamics are often amplified, and high-frequency monitoring can uncover critical diel fluctuations that are missed by traditional low-frequency sampling

methods.

The aim of this study was to investigate changes in water quality, ecosystem metabolism, and nitrogen uptake during the extreme summer low flow of 2018 in a 6th-order lowland agricultural stream in Central Germany. Using high-frequency in *situ* water quality sensor data collected at 15-minute intervals at upstream and downstream stations along the Lower Bode reach, we analyzed conditions during 2018 and compared them to summers from 2014-2017.

Specifically, we aimed to 1) assess whether the 2018 extreme summer low flow significantly altered various water quality indicators, 2) investigate how ecosystem metabolism and autotrophic uptake pathways responded to the 2018 extreme low flow, 3) determine whether the stream became more efficient in nitrogen removal and processing during the 2018 extreme summer low flow. This study offers valuable evidence for future water quality management strategies in the context of increasing extreme summer low-flow events driven by climate change.

## 2 Methods

### 2.1 Study site

The Bode River is located in Saxony-Anhalt, central Germany (Fig. 1). It is 169-km long and drains a catchment area of 3270 km², originating from the Harz Mountains and flowing into the Saale River. The Bode Catchment is one of the most intensively monitored areas within the Terrestrial Environmental Observatories (TERENO) network

a research initiative operated by the Helmholtz Centre for Environmental Research (UFZ) (http://www.tereno.net, last access: 27 Nov 2024).

This study focuses on the Lower Bode, a 6th-order lowland agricultural stream reach characterized by a flat river slope of 0.0004 and predominantly rectangular or trapezoidal cross sections. Over the past century, the original meandering reach was largely straightened or re-routed artificially. The reach is only partly shaded by lined

deciduous trees on the riverbanks (Fig. 1). The reach has an average water depth of 1.5 m and a width of 20 m, with a multi-year mean discharge of 12.2 m³ s⁻¹ at the downstream end (https://hochwasservorhersage.sachsen-anhalt.de/messwerte/durchfluss/, last access: 27 Nov 2024). The open canopy and low shading allow for high solar





irradiance at the water surface, fostering the development of phytoplankton and benthic algae, including large mats of periphyton and macrophytes. Land use in the catchment is dominated by intensive agriculture, with urban areas playing a minor role. The Lower Bode is equipped with high-frequency discharge and water quality monitoring stations at both ends of the study reach. These stations provide continuous data that form the basis of this study, offering a detailed understanding of the river's hydrological and ecological dynamics.

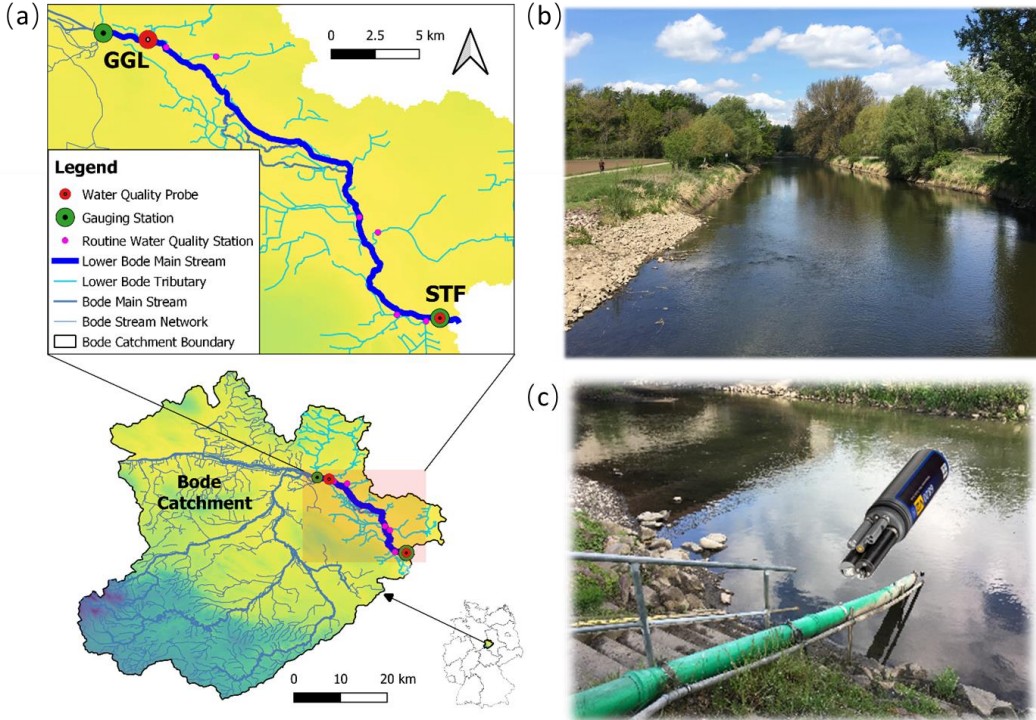

**Figure 1: (a) Location of the Lower Bode in the Bode Catchment and within Germany and locations of gauging stations and water quality probes within the study reach, (b) photo of the study reach, and (c) photo of the STF high-frequency water quality monitoring station.**

**2.2 Hydrology and extreme summer low-flow condition**

For the Lower Bode, discharge data were obtained from two active gauging stations, namely Hadmersleben (HAD) upstream and Staßfurt (STF) downstream. These stations are managed by the State Agency for Flood Protection and Water Management of Saxony-Anhalt (LHW). Daily flow percentiles were calculated from flow duration curves derived from long-term discharge data spanning 88 water years for HAD and 30 years for STF (Fig.2, https://hochwasservorhersage.sachsen-anhalt.de/messwerte/durchfluss/, last access: 27 Nov 2024).

In 2018, Central Germany, including the Bode catchment, experienced a severe and prolonged drought. For the Lower Bode, this resulted in an extended low-flow period lasting from summer through autumn. This study focuses on two summer months 2018 namely July and August, which are statistically identified as extreme low flow (hereafter referred to as ExLF). During this period, median daily flows at HAD (2.48 m$^3$ s$^{-1}$) and at STF (2.47 m$^3$ s$^{-1}$)



were below the 1[th] percentile of their respective long-term records (Fig. 2). The discharge data confirm that flow in the study reach was dominated by upstream inflow from the upper catchment (Huang et al., 2022). Additionally, visual inspections by LHW confirmed that tributaries contributing to the reach had dried up during ExLF. The nearly identical median flows at HAD and STF indicate an absence of significant tributary inflows during this period.

To evaluate water quality responses to ExLF, we compared them to those observed 'normal' summer low-flow conditions. For this reference period (hereinafter referred to as LF), we selected the same months, July and August, from 2014 to 2017. To ensure consistency, we excluded data from periods when discharge exceeded the long-term monthly mean discharge (MQ) for July (6.09 $m^3$ $s^{-1}$) and August (6.33 $m^3$ $s^{-1}$) at STF (Fig. 2; MQ values from https://hochwasservorhersage.sachsen-anhalt.de/messwerte/durchfluss/, last access: 27 Nov 2024). The median daily flow at STF during LF (3.81 $m^3$ $s^{-1}$, Fig. 2) was still below the 10[th] percentile of the long-term records but significantly higher than the ExLF level (2.47 $m^3$ $s^{-1}$). High-frequency discharge data were collected at 15-minute intervals at both HAD and STF throughout the study period.

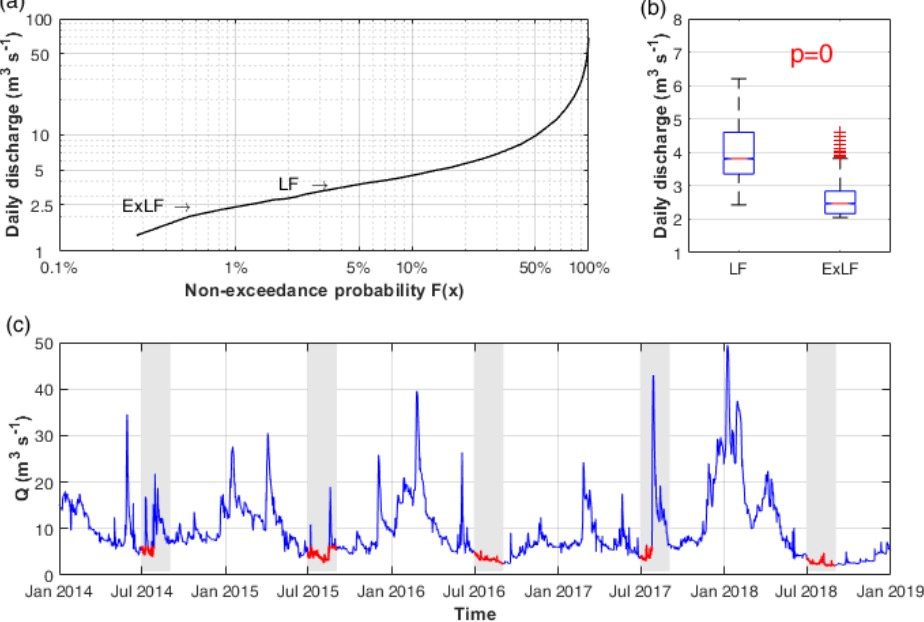

**Figure 2: (a) Flow-duration curve at STF with median discharge values during LF and ExLF shown with arrows. (b) Boxplot of discharge at STF during LF and ExLF. (c) Discharge time series in the study periods (grey bars) at STF; The discharge values no higher than the MQ values of July and August were kept in the study period for comparison (red lines).**



### 2.3 High-frequency water quality measurement

High-frequency water quality data were collected at 15-minute intervals at two monitoring stations: Groß Germersleben (GGL), located 2.7 km downstream of HAD, and Staßfurt (STF), during the ExLF and LF periods

defined above (Fig. 1). Water temperature (WT), dissolved oxygen (DO), and chlorophyll-a (*Chl* a) were measured using a YSI 610 multiparameter probe. Nitrate ($NO_3^-$) concentrations were monitored with a TRIOS ProPS-UV sensor featuring a 10 mm optical path length. The nitrate sensor performed automated self-cleaning using air pressure before each measurement. Additionally, all sensors were maintained monthly, which included manual cleaning and calibration to ensure data accuracy.

Data post-processing involved rigorous screening to remove outliers. Grubb's test was applied with a moving window approach to identify and eliminate anomalies. The nitrate sensor readings were further corrected using linear regression against laboratory-analyzed grab sample data to ensure precision.

Under summer low-flow conditions, DO and Chl *a* concentrations exhibited clear diurnal patterns at both stations (Fig. 4), consistent with primary production and respiration cycles. A custom MATLAB script was used to extract

key metrics such as daily maximum, minimum, and delta values for DO and Chl *a* concentrations over the study period.

### 2.4 Ecosystem metabolism

Single-station open channel diel DO models with high-frequency in *situ* DO measurements were used to quantify ecosystem metabolisms, including areal rates of gross primary productivity (GPP; $gO_2$ m$^{-2}$ d$^{-1}$), ecosystem

respiration (ER; $gO_2$ m$^{-2}$ d$^{-1}$), net ecosystem productivity (NEP; $gO_2$ m$^{-2}$ d$^{-1}$), and the ratio of GPP to ER (GPP:ER; dimensionless) for each day (Rode et al., 2016a). The modeling approach integrates changes in DO concentrations over time, accounting for oxygen production by autotrophs during daylight, oxygen consumption by biota throughout the day and night, and reaeration driven by gas exchange with the atmosphere. The reaeration rate is estimated using the O'Connor-Dobbins formula, which is suitable for slow-flowing streams (Chapra, 2008). The

independent variables in the formula, including water depth and flow velocity, are available from the gauging station readings and hydrodynamic modeling results (Huang et al., 2022). DO saturation level was determined from the measurement of water temperature, salinity, and barometric pressure (APHA, 1998).

Phytoplankton and benthic algae co-exist and determine the whole stream GPP of the Lower Bode (Huang et al., 2022). To distinguish their contributions, we first calculated the areal rate of phytoplankton GPP ($GPP_P$) with the

equation as follows:

$$GPP_P = G_P \times C_{PhyC} \times ROC \times z \tag{1}$$

where $G_P$ is phytoplankton growth rate (d$^{-1}$); $C_{PhyC}$ represents the concentration of phytoplankton biomass carbon (mg C L$^{-1}$); $ROC$ represents oxygen to carbon ratio of 2.67 ($gO_2$/gC); $z$ donates the stream depth (m). $G_P$ was taken from our previous water quality model applied to the same reach (Huang et al., 2022), which nicely captured the phytoplankton growth between GGL and STF and reproduced both seasonal and diurnal patterns of Chl *a* (Fig. S1).

Then, the GPP for benthic algae ($GPP_B$) was calculated by subtracting the $GPP_P$ from whole stream GPP. Note that



here benthic algae refer to the whole primary producer community in benthic habitats, including both periphyton and macrophytes.

**2.5 NO$_3^-$ Uptake**

The daily areal rate of net NO$_3^-$ uptake ($U_{NET-NO_3^-}$, in mgN m$^{-2}$ d$^{-1}$) for the study reach was calculated using a mass

balance approach as follows:

$$U_{NET-NO_3^-} = \frac{C_{US(t)} \times Q_{US(t)} + \sum C_{TR(t)} \times Q_{TR(t)} - C_{DS(t+t_{Flow})} \times Q_{DS(t+t_{Flow})}}{A_{Reach}} \times 86.4 \quad (2)$$

where the $C_{US(t)}$ and $Q_{US(t)}$ represent the NO$_3^-$ concentration (mgN L$^{-1}$) and discharge (m$^3$ s$^{-1}$) at upstream stations GGL at time $t$; the $\sum C_{TR(t)} Q_{TR(t)}$ denote summed NO$_3^-$ loadings from small tributaries; $C_{DS(t+t_{Flow})}$ and $Q_{DS(t+t_{Flow})}$ represent NO$_3^-$ concentration (mgN L$^{-1}$) and discharge (m$^3$ s$^{-1}$) at downstream station STF at time $t+t_{Flow}$; $A_{Reach}$ is the stream bottom area of the study reach (m$^2$); 86.4 is the unit converter from gN m$^{-2}$ s$^{-1}$ to mgN m$^{-2}$

d$^{-1}$. The water travel time ($t_{Flow}$, in d) is between the upstream and the downstream stations was derived from hydrodynamic modeling results (Huang et al., 2022; Fig. S3). The discharge from tributaries ($Q_{TR(t)}$) was estimated using the specific discharge method, referencing daily discharge measurements at Geesgraben, a gauging station ~0.5 km upstream of HAD. Daily nitrate concentrations in tributaries ($C_{TR(t)}$) were derived from discharge-concentration($C$-$Q$) linear regressions. A detailed description of tributary discharge and nitrate estimations is

available in Huang et al. (2022). Tributary travel time delays were excluded, as tributary contributions were minor during LF compared to the upstream loading and even negligible during ExLF due to dry channels. Stable flow condition at summer low flows, especially during ExLF, increases the reliability of the mass balance method, as the system approximates a quasi-steady state rather than experiencing transient dynamic characteristic of peak flow events. The stream bottom area ($A_{Reach}$) was calculated using 413 cross-section measurements, revealing mostly

rectangular profiles along the Lower Bode. The calculations of $U_{Net-NO_3^-}$ were performed at 15-min intervals and aggregated to daily values.

The percentage net NO$_3^-$ uptake relative to total input loading ($Perc_{NET-NO_3^-}$, in %) was calculated as:

$$Perc_{NET-NO_3^-} = \frac{U_{Net-NO_3^-} \times A_{Reach}}{I} \times 100\% \quad (3)$$

where $I$ represents the total NO$_3^-$ input loadings (sum of loadings from upstream and tributaries) in mg d$^{-1}$.

**2.6 Statistical analysis**

To evaluate differences in water quality and stream process variables between the reference summer low flow (LF) and extreme summer low flow (ExLF) periods at GGL and STF stations in the Lower Bode, we employed the Kruskal–Wallis test (Kruskal and Wallis, 1952). This non-parametric test is well-suited for comparing datasets that do not meet the assumptions of normality or homogeneity of variance. A total of 24 variables were analyzed including: **Hydrological variables:** discharge (Q), travel time (t$_T$); **Physical parameters:** solar radiation, water

temperature (WT); **DO dynamics:** DO concentration, daily DO maximum (DO$_{max}$), daily DO minimum (DO$_{min}$), daily DO delta (DO$_\Delta$), daily DO saturation (DO$_S$), daily DO deficit (DO$_D$); **Chl _a_ metrics:** Chl _a_ concentration,





daily Chl $a$ maximum (Chl $a_{max}$), daily Chl $a$ minimum (Chl $a_{min}$), daily Chl $a$ delta (Chl $a_\Delta$), Chl $a$ accumulation (Chl $a_{ACC}$); **Nitrate and metabolic rates:** $NO_3^-$ concentration, GPP, ER, NEP, P:R, phytoplankton GPP (GPP$_P$), benthic algae GPP (GPP$_B$), areal $NO_3^-$ net uptake rate ($U_{NET-NO_3^-}$), and percentage $NO_3^-$ net uptake ($Perc_{NET-NO_3^-}$).

Statistical significance for all tests was defined at a threshold of $p \leq 0.01$. The analysis was implemented using Matlab2020a.

## 3 Results and Discussion

### 3.1 Increased water temperature during ExLF

The Kruskal–Wallis test revealed significantly higher water temperatures ($p < 0.01$) during ExLF compared to LF at
both stations, GGL and STF (Fig. 3a, Table 1). Median values during ExLF were elevated by 0.43 °C at GGL and 0.90 °C at STF relative to LF (Table 1). Notably, the water temperature at STF exceeded the ecological threshold of 25 °C —critical for the survival of many aquatic organisms—on almost 8.66 days during the 62-days ExLF period. By contrast, during the four two-month LF periods, this threshold was surpassed in total only 4.71 days (Table 1).

The increase in WT during ExLF is expected, as extreme low-flow conditions can lead to extended water residence
times and reduced water volumes, thereby lowering the thermal capacity of the river (Lake, 2003; Mosley, 2015). Such temperature elevations during drought conditions are widely reported in previous studies (Ahmadi and Moradkhani, 2019; Mosley et al., 2012; Riis et al., 2017). For instance, Addy et al. (2018) observed water temperature increases of 2.1–2.5 °C during the summer drought of 2016 compared to comparable months in a reference year using high-frequency sensors.

The warming effect at STF during ExLF was significantly more pronounced than at GGL (Table 1). This disparity can be attributed to differences in riparian canopy cover and shading. The 6th-order reach upstream of STF is largely straightened and poorly shaded (LHW, 2012). At STF, the high-frequency sensors were installed in a channelized urban reach with virtually no canopy, exposing the site to direct solar radiation. In contrast, upstream of GGL, the reach has a higher canopy cover. The low shading and open canopy of the Lower Bode allow high irradiance at the
water surface and thus the stronger warming impact of ExLF on the WT at STF than at GGL. The significant difference in WT responses to ExLF at GGL and STF suggests the importance of riparian canopy shading to mitigate the warming impact of extreme low flow on streams. In a study examining WT across stream orders in a nested network, Hosen et al. (2019) reported that drought had a greater warming impact on water temperature in higher-order rivers than in canopy creeks, also noting the importance of canopy cover on WT responses.

Temperature increases also accelerate reaction rates in natural waters, as described by the simplified Arrhenius equation (Chapra, 2008):

$$\frac{k(T_2)}{k(T_1)} = \theta^{T_2 - T_1} \tag{4}$$

where $T_2$ and $T_1$ are temperatures in °C, and θ (typically 1.047–1.1) represents a temperature correction coefficient that varies with the type of reaction (Chapra, 2008). Assuming θ=1.08, the biological reaction rate during ExLF is




estimated to increase by approximately 7.1% at STF and 3.3% at GGL compared to LF. Thus, any reaction rates
exceeding these increases may suggest additional drivers beyond temperature.

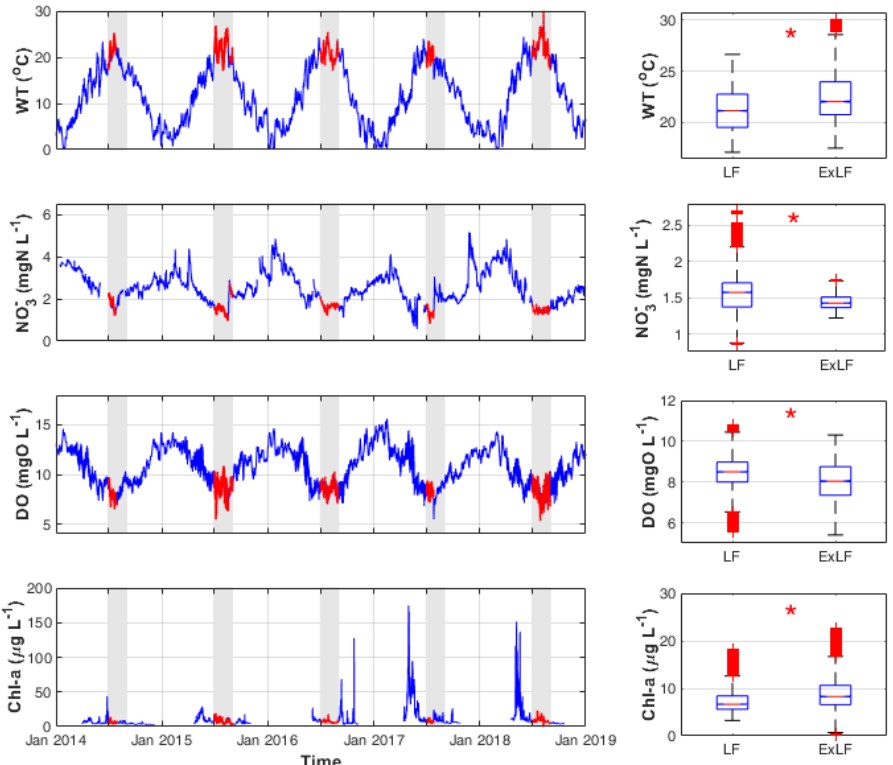

**Figure 3: Time series (left panel) of water temperature (WT), NO₃⁻, DO, and Chl *a* concentration measured at 15-min intervals at STF in the study periods (grey bars) and boxplots of the variables during LF and ExLF (right panel). * shows the test variable is significantly different according to the Kruskal–Wallis one-way analysis of variance; *p* value < 0.01.**





**Table 1 Kruskal–Wallis one-way analysis of variance for water quality variables and ecosystem processes at LF and ExLF.**

| Variable | Unit | LF | ExLF | p |
|---|---|---|---|---|
| Q_GGL | $m^3\ s^{-1}$ | 3.86* | 2.48 | 0.00 |
| Q_STF | $m^3\ s^{-1}$ | 3.81* | 2.47 | 0.00 |
| WT_GGL | °C | 19.07 | 19.49* | 0.00 |
| WT_STF | °C | 21.13 | 22.02* | 0.00 |
| NO₃⁻_GGL | $mg\ N\ L^{-1}$ | 1.61* | 1.55 | 0.00 |
| NO₃⁻_STF | $mg\ N\ L^{-1}$ | 1.58* | 1.43 | 0.00 |
| tr | day | 1.46 | 2.11* | 0.00 |
| Solar Radiation | | 103 | 114 | 0.01 |
| DO_GGL | $mg\ L^{-1}$ | 8.75* | 8.45 | 0.00 |
| DO_STF | $mg\ L^{-1}$ | 8.50* | 8.04 | 0.00 |
| DO$_{max}$_GGL | $mg\ L^{-1}$ | 10.00 | 10.45* | 0.00 |
| DO$_{max}$_STF | $mg\ L^{-1}$ | 9.04 | 9.32 | 0.11 |
| DO$_{min}$_GGL | $mg\ L^{-1}$ | 7.72* | 7.01 | 0.00 |
| DO$_{min}$_STF | $mg\ L^{-1}$ | 7.90* | 7.17 | 0.00 |
| DO$_A$_GGL | $mg\ L^{-1}$ | 2.36 | 3.54* | 0.00 |
| DO$_A$_STF | $mg\ L^{-1}$ | 1.05 | 1.78* | 0.00 |
| DO$_S$_GGL | $mg\ L^{-1}$ | 9.31 | 9.20 | 0.00 |
| DO$_S$_STF | $mg\ L^{-1}$ | 8.93* | 8.74 | 0.00 |
| DO$_D$_GGL | $mg\ L^{-1}$ | 0.48 | 0.60 | 0.14 |
| DO$_D$_STF | $mg\ L^{-1}$ | 0.49 | 0.64 | 0.03 |
| Chl $a$_GGL | $µg\ L^{-1}$ | 3.93* | 3.62 | 0.00 |
| Chl $a$_STF | $µg\ L^{-1}$ | 6.65 | 8.27* | 0.00 |
| Chl $a_{max}$_GGL | $µg\ L^{-1}$ | 4.67 | 4.47 | 0.23 |
| Chl $a_{max}$_STF | $µg\ L^{-1}$ | 7.60 | 9.87* | 0.00 |
| Chl $a_{min}$_GGL | $µg\ L^{-1}$ | 3.73* | 3.28 | 0.00 |
| Chl $a_{min}$_STF | $µg\ L^{-1}$ | 6.05 | 6.63* | 0.00 |
| Chl $a_A$_GGL | $µg\ L^{-1}$ | 0.65 | 1.14* | 0.00 |
| Chl $a_A$_STF | $µg\ L^{-1}$ | 1.63 | 2.82* | 0.00 |
| Chl $a_{ACC}$ | $µg\ L^{-1}$ | 2.97 | 4.64* | 0.00 |
| GPP | $g\ O_2\ m^{-2}\ d^{-1}$ | 1.83 | 2.67* | 0.00 |
| ER | $g\ O_2\ m^{-2}\ d^{-1}$ | 2.48 | 3.05* | 0.00 |
| NEP | $g\ O_2\ m^{-2}\ d^{-1}$ | -0.55 | -0.46 | 0.45 |
| GPP:ER | -- | 0.80 | 0.84 | 0.04 |
| GPP_Phy | $g\ O_2\ m^{-2}\ d^{-1}$ | 0.38 | 0.46 | 0.30 |
| $U_{NET-NO_3^-}$ | $mg\ N\ m^{-2}\ d^{-1}$ | 99.79 | 101.07 | 0.70 |
| $Perc_{NET-NO_3^-}$ | % | 6.83 | 11.69* | 0.00 |

The value with a * is statistically significantly higher ($p < 0.01$).




### 3.2 DO responses

DO concentrations during ExLF were significantly lower ($p < 0.01$) compared to LF at both GGL and STF stations (Fig. 3c, Table 1). Median DO concentrations during ExLF decreased by 0.29 mg L$^{-1}$ at GGL and 0.46 mg L$^{-1}$ at STF relative to LF.

The DO concentrations decreased at ExLF, as reported in many other studies (Ahmadi and Moradkhani, 2019; van Vliet and Zwolsman, 2008). DO displayed a diurnal pattern during both LF and ExLF (Fig. 4), with this pattern strongly amplified during ExLF. Median daily maximum DO concentrations increased significantly at GGL by 0.45 mg L$^{-1}$ at GGL ($p < 0.01$) and non-significantly at STF 0.28 mg L$^{-1}$ ($p = 0.11$). Conversely, daily minimum DO concentrations dropped significantly at both sites, by 0.71 mg L$^{-1}$ at GGL ($p < 0.01$) and 0.73 mg L$^{-1}$ at STF ($p < 0.01$). Consequently, daily DO deltas (maximum

minus minimum) rose significantly, increasing by 50% at GGL and 69% at STF during ExLF (Table 1). In addition, the medians of daily DO saturation level decreased by 0.11 and 0.19 mg L$^{-1}$ at GGL and at STF. The medians of daily DO deficit increased by 0.12 and 0.15 mg L$^{-1}$ at GGL and at STF, respectively. The close matching values for the decrease in DO saturation and the increase in DO deficit indicated that the decrease in DO levels during ExLF was mainly influenced by the rising water temperature. Still, it is worth noting that the minimum DO concentration during ExLF did not fall below the

ecological threshold of 5 mg L$^{-1}$.

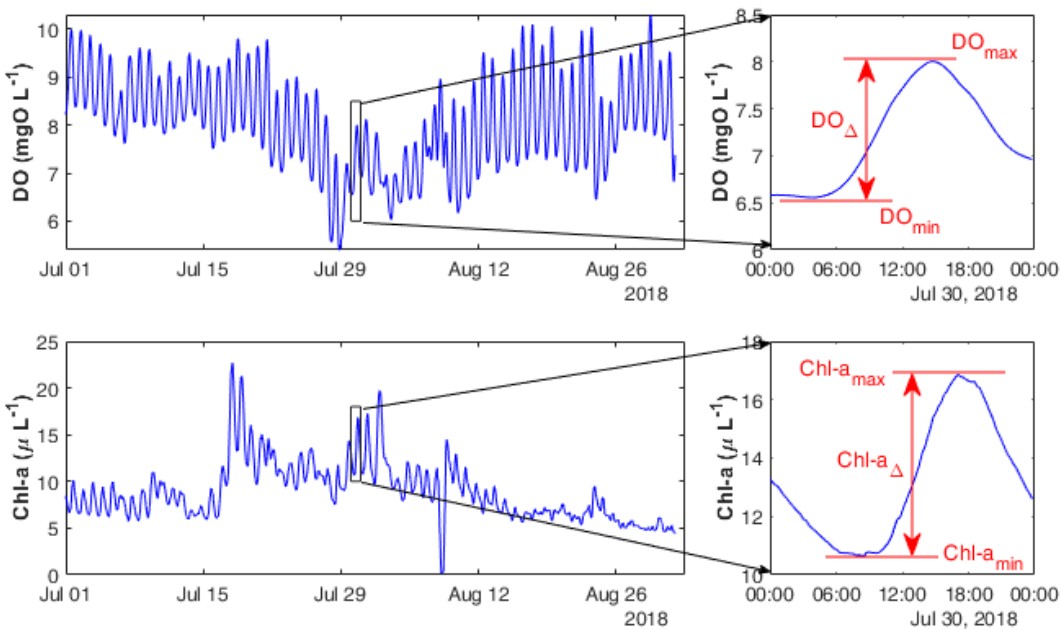

**Figure 4: DO and Chl *a* concentrations at the STF station during the extreme summer low flow period of 2018 and their diurnal variations with the DO$_{max}$, DO$_{min}$, DO$_{\Delta}$, Chl *a*$_{max}$, Chl *a*$_{min}$, and DO$_{\Delta}$ values annotated in red.**



As a lowland agricultural stream, the Lower Bode is not heavily influenced by point sources. Instead, its DO balance is
governed by ecosystem processes such as photosynthesis and respiration. Photosynthesis increases DO concentrations during
daylight hours, while respiration consumes oxygen throughout the day (Odum, 1956). The increased in daily DO max and
decreased daily min values were signs of elevated gross primary productivity and a synchronized increase in ecosystem
respiration. The amplified daily DO deltas further indicate intensified ecosystem metabolism rates during ExLF.
Additionally, the decrease in DO daily minimum at both sites greatly exceeded the decrease in their daily averages, which is
likely due to enhanced respiration during ExLF. This phenomenon implies that even if the river is not affected by point
sources, its DO levels can be adversely affected by the intensive metabolic processes during ExLF.

Depending on the timing of the day sampled, completely different conclusions can be drawn about changes in DO
concentrations in the Lower Bode during ExLF. For example, a conclusion from the afternoon DO sampling might be that
DO levels during ExLF increased significantly, whereas sampling only in the early morning around sunrise can lead to the
exact opposite conclusion. Thus, the timing of the discrete sampling may alter the conclusion about the DO impacts and
further the process interpretation, which might lead to incomplete or even misleading findings about the impacts during
ExLF. This variability highlights the risk of incomplete or misleading interpretations from discrete sampling alone. In
addition, the time of day when the lowest DO levels typically occur pre-sunrise is rarely covered by manual grab sampling.
Using high-frequency DO measurements captures critical DO variations and provides a comprehensive understanding of the
DO responses, thereby facilitating timely and effective water quality management and interventions under extreme low-flow
conditions.

### 3.3 Slight but significant increase in Chl $a$ levels

Chl $a$ concentrations (a proxy of phytoplankton biomass) at GGL were significantly lower ($p < 0.01$, Table 1) during ExLF
compared with LF, while its concentrations at STF were significantly higher ($p < 0.01$, Table 1). Therefore, the difference in
Chl $a$ concentrations (Chl $a_{ACC}$) between the two stations increased significantly ($p < 0.01$), with the median increasing from
2.97 µg L$^{-1}$ during LF to 4.64 µg L$^{-1}$ during ExLF (Table 1), which indicates that phytoplankton accumulated more in the
Lower Bode during ExLF than LF. The Chl $a_{max}$ and Chl $a_{min}$ concentrations decreased at GGL during ExLF. In contrast, the
Chl $a_{max}$ and Chl $a_{min}$ concentrations both increased at STF during ExLF. The Chl $a_\Delta$ increased significantly ($p < 0.01$) at
both stations. Specifically, the Chl $a_\Delta$ values increased from 0.65 µg L$^{-1}$ to 1.14 µg L$^{-1}$ at GGL and from 1.63 µg L$^{-1}$ to 2.82
µg L$^{-1}$ at STF.

Diatoms were identified as the dominant phytoplankton taxa in the Lower Bode (Kamjunke et al., 2015). Our previous study
found that the average phytoplankton Chl $a$ levels peaked when the water temperature ranged between 10-14 °C (Huang et
al., 2022), aligning with the optimal growth temperature of diatoms (Chapra, 2008).Thus, the temperature increase during
ExLF did not favor the phytoplankton growth in the Lower Bode. However, the longer residence time during ExLF
facilitated the Chl $a$ accumulation within the reach. Despite this, the Chl $a$ concentrations at both stations remained low level
within a desirable range of mesotrophic status, and no severe phytoplankton bloom was formed during ExLF. This may be




due to the two factors: firstly, the Chl *a* levels from the upstream inflow did not increase during ExLF, and secondly, elevated temperatures constrained the growth rate of planktonic diatoms. Flow-induced shifts in stream communities were widely found, e.g., a regime cyanobacteria shift (Chapra et al., 2017; Mitrovic et al., 2010; Rosero-López et al., 2022).

Cyanobacteria development can benefit from increased water temperature and longer residence time during extreme low flows, so that it eventually replaces "the good algae" and dominates the algae community (Chapra et al., 2017). However, our results showed that ExLF 2018 did not push the system to a tipping point where the phytoplankton taxa shift occurred. Since 2018, consecutive summer droughts have impacted the Bode catchment, warranting further studies on potential harmful algal blooms and phytoplankton community shifts under persistent extreme low-flow conditions.

A significant diurnal pattern in Chl *a* concentrations was observed during the summer low flows in the Lower Bode (Fig. 4), which has been reproduced by the water quality modeling results in Huang et al. (2022). The modeling results suggested that increased the phytoplankton growth rates led to higher overall Chl *a* concentrations and greater diurnal fluctuations (Fig. S2). The significantly larger diurnal deltas of Chl *a* measurement during ExLF suggested higher primary production by phytoplankton compared to LF, as evidenced by the greater Chl *a* accumulation during ExLF (Table 1).

Few studies have examined Chl *a* diel fluctuations compared with DO variations (Pathak et al., 2021; Poulin et al., 2018). This is likely due to two reasons: (1) high-frequency Chl *a* measurements are less often conducted in running waters compared to DO and (2) Chl *a* fluorescence, as a proxy of phytoplankton biomass, is often biased by non-photochemical quenching (NPQ). NPQ, a phenomenon of photoprotection processes triggered by increased irradiance, can cause midday reductions in Chl a signals, potentially leading to misinterpretations of biomass declines (Lucius et al., 2020). Many studies

report NPQ-induced reductions in Chl a signals around solar noon (Carberry et al., 2019; Hamilton et al., 2015). However, our results demonstrate a diurnal Chl *a* in the Lower Bode during summer low flows, unaffected by NPQ. Similarly, Pathak et al. (2021) observed diurnal Chl *a* fluctuations in the River Thames using hourly measurements, which were successfully reproduced by a water quality model. More high-frequency Chl *a* measurements in running waters are needed as their diurnal fluctuations may provide new insights into phytoplankton productivity dynamics.

### 325 **3.4 Decreased $NO_3^-$ concentration with higher spatial heterogeneity**

$NO_3^-$ concentrations at ExLF were significantly lower ($p < 0.01$) compared to LF at both stations (Fig. 3b, Table 1). The median values of $NO_3^-$ concentration at ExLF decreased by 0.06 mg N $L^{-1}$ at GGL and 0.15 mg N $L^{-1}$ at SFT compared to LF (Table 1). Additionally, the difference in $NO_3^-$ concentrations between the two stations increased from 0.03 mg N $L^{-1}$ during LF to 0.12 mg N $L^{-1}$ during ExLF, indicating a significant enhancement in spatial heterogeneity of $NO_3^-$ levels between

upstream and downstream under ExLF conditions.

The Lower Bode, a typical agricultural lowland stream, receives $NO_3^-$ inputs primarily from agricultural surface runoff, interflow, and groundwater (Zhou et al., 2022). Across both stations and tributaries within the study reach, the concentration-discharge (C-Q) relationship exhibited enrichment responses with positive slopes (Huang et al., 2022). These results suggested that $NO_3^-$ is transport/mobile limited in the Bode catchment (Wollheim et al., 2018). The observed $NO_3^-$ reduction

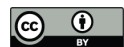



at both sites during ExLF resulted from a weakened hydrological connection between the catchment and the river. Specifically, the export of $NO_3^-$ loadings from agricultural runoff declined significantly as discharge in decreased the Bode catchment during ExLF (Zhou et al., 2022). Some drainage tributaries within the study reach even dried up during ExLF, drastically reducing the catchment-to-stream $NO_3^-$ load contribution. Furthermore, the Lower Bode reach lost 0.57% of its inflow during ExLF, indicating an absence of groundwater inputs, which are typically rich in $NO_3^-$ concentrations. Beyond

reduced lateral $NO_3^-$ loading, increased instream processing could have further amplified spatial heterogeneity between upstream and downstream. Consequently, the spatial disparity in $NO_3^-$ concentrations was significantly greater during ExLF compared to LF, aligning with findings by Hensley et al. (2019). Similar reductions in $NO_3^-$ concentrations during droughts have been reported in many rivers and streams, often attributed to diminished catchment inputs and enhanced in-stream retention (Caruso, 2001; Mosley et al., 2012; Muchmore and Dziegielewski, 1983).

### 345   3.5  Less heterotrophy primarily attributed to increased benthic algae GPP

The areal GPP rate in the Lower Bode increased significantly by 46% ($p < 0.01$), with the median values rising from 1.83 g $O_2$ $m^{-2}$ $day^{-1}$ during LF to 2.67 g $O_2$ $m^{-2}$ $day^{-1}$ during ExLF (Table 1). The areal ER rate (negative values indicate oxygen consumption) also increased, from -2.48 g $O_2$ $m^{-2}$ $day^{-1}$ during LF to -3.05 g $O_2$ $m^{-2}$ $day^{-1}$ during ExLF (Table 1). However, the increase in GPP during ExLF exceeded that of ER, resulting in an areal NEP rate closer to zero (Table 1). This indicates

that although the system remained heterotrophic during ExLF, it was less heterotrophic than during LF. The median areal rate of $GPP_P$ increased from 0.38 $O_2$ $m^{-2}$ $day^{-1}$ during LF to 0.46 g $O_2$ $m^{-2}$ $day^{-1}$ during ExLF, while that of $GPP_B$ increased from 1.45 g $O_2$ $m^{-2}$ $day^{-1}$ during LF to 2.21 g $O_2$ $m^{-2}$ $day^{-1}$ during ExLF (Table 1).

During ExLF, the Lower Bode exhibited greater GPP, driven by enhanced productivity from both phytoplankton and benthic algae. However, the dominant contributor to this increase was benthic algae, accounting for 95% of the total GPP

enhancement. Extreme low flows alter river hydraulics, water temperature, and nutrient and light availability, all of which influence riverine GPP. Phytoplankton growth and $GPP_P$ benefited from the longer residence time during ExLF (Table 1, Section 3.3), but elevated water temperatures constrained the magnitude of the GPP enhancement by diatom-dominated phytoplankton (Section 3.3). In contrast, macrophytes in the benthic compartment, particularly *Potamogeton pectinatus*, the dominant species of macrophytes in the Lower Bode (LHW, 2012), are more tolerant of high water temperature. This species

exhibits a broad photosynthetic response across temperatures ranges, which peak net photosynthesis occurring at an optimal temperature of 30 °C (Madsen and Adams, 1989). Consequently, the elevated water temperature during ExLF likely promoted $GPP_B$. However, according to the calculations using the simplified Arrhenius equation (Section 3.1), the temperature-driven increase is about 7.1% during ExLF in the Lower Bode. This increase alone does not fully explain the total percentage increase in $GPP_B$ during ExLF, i.e., 53%. which suggests additional factors are at play.

One key factor was increased light availability for benthic algae. Two primary mechanisms contributed to this: (1) less cloud cover during drought and the low canopy cover of the study reach allowed higher solar radiation to reach the water surface (Table 1). (2) Lower water depth and velocity, coupled with reduced sediment transport and turbidity, allowed more light





penetration to the benthic compartment. Both greater solar radiation and light penetration likely contributed to increased light availability for benthic algae during ExLF, which enabled higher GPP$_B$ (Mosley, 2015). A study on drought impacts on

GPP across stream orders pointed out that the reduction of turbidity below 10 FNU still led to increased GPP, indicating that the primary productivity was very sensitive to the light response (Hosen et al., 2019). The high temperature tolerance of benthic algae allowed them to take advantage of this increased light availability to facilitate its productivity. Thus, increased temperatures and light availability synergistically increased GPP$_B$ during ExLF in the Lower Bode. Additionally, N did not appear to be a limiting factor for GPP in the Lower Bode, as GPP continued to increase despite lower N concentrations

during ExLF.

Enhanced benthic algal production during droughts has also been observed in previous studies. For example, Caruso (2001) observed benthic algae blooms during drought in lowland agricultural catchments with low water velocities and minimal shade. Although many studies have reported GPP increases under drought conditions (Addy et al., 2018; Hosen et al., 2019), others have found the opposite, particularly in temporal headwater streams, where network contraction during drought limits

primary production (Amalfitano et al., 2008; Crawford et al., 2017). Ultimately, whether extreme low flow enhances or limits primary production in a stream depends on the local environmental conditions, such as canopy cover, dominant primary producer taxonomy, and their optimal growth conditions.

In addition to increased GPP rate, the Lower Bode also exhibited high ER rate during ExLF. However, the environmental conditions during ExLF stimulated GPP more than ER. Thus, the system remained heterotrophic during ExLF, but to a lesser

extent. Similar findings were reported by Hosen et al. (2019), who observed that GPP increased more than ER during drought, occasionally leading to temporary autotrophy—an otherwise rare occurrence in the typically light-limited heterotrophic Connecticut River. Hensley et al. (2019) also found reduced heterotrophy with declining flows in the Lower Santa Fe River. Although ER is generally more responsive to temperature increases than GPP (Allen et al., 2005), temperatures alone can also not fully explain the 23% increase in ER during ExLF, as calculations using the simplified

Arrhenius equation suggest that biological activity should have increased by only 7.1%. Another likely factor contributing to the elevated ER during ExLF is the increased release of labile autochthonous dissolved organic matter (DOM) by algae, which has been shown to stimulate ER (Bertilsson and Jones, 2003).

Ultimately, flow-mediated changes during ExLF led to stronger nonlinear responses in GPP than ER In the open-canopy Lower Bode reach. This discrepancy likely arises from the differing sensitivities of GPP and ER to increased light

availability during extreme low flows (Hensley et al., 2019; Hosen et al., 2019). Because stream metabolism can exert control within the stream on nutrient and carbon cycling, such changes in metabolism towards less heterotrophy or even autotrophy in rivers have broad implications for carbon cycling and water quality in aquatic ecosystems, particularly as extreme summer low flows become more frequent and severe in the future.





### 3.6 Non-significant change in areal NO₃⁻ net uptake rate

The Kruskal-Wallis-Test results showed that the median of $U_{NET-NO_3^-}$ during ExLF (101.07 mgN m$^{-2}$ d$^{-1}$) was slightly higher than during LF (99.79 mgN m$^{-2}$ d$^{-1}$), but the difference was not statistically significantly ($p$ = 0.70, Table 1). However, the $Perc_{NET-NO_3^-}$ during ExLF increased significantly, with a median of 11.7% compared to 6.8% during LF.

In nitrate-rich streams, nitrogen availability does not limit autotrophic N demand (Covino et al., 2018). Therefore, given the significant 46% increase in areal GPP during ExLF, it is likely that gross NO₃⁻ uptake rates also increased. However, despite

this, $U_{net}$–NO₃⁻ did not show a significant change during ExLF. This discrepancy between gross and net NO₃⁻ uptake may be attributed to the dominant contribution of benthic algae (GPP$_B$) to the overall GPP increase at ExLF (Section 3.5). The simultaneous processes of nitrogen assimilation and excretion by benthic algae may have influenced net uptake rates (Huang et al., 2022). Von Schiller et al. (2015) described net N uptake as a quasi-equilibrium process over short timeframes, as nitrogen continuously cycles through algae and is subsequently released back into the water column. During ExLF, increased

N assimilation by benthic algae would also result in greater N release. These released nitrogen forms are then remineralized and nitrified, regenerating NO₃⁻ and effectively decoupling net NO₃⁻ uptake rates from gross uptake dynamics.

Although the areal net NO₃⁻ uptake rate did not increase significantly, the elevated GPP during ExLF likely accelerated internal processing and recycling of NO₃⁻ within the Lower Bode (Bruesewitz et al., 2013). NH₄⁺ can be actively involved in the N internal cycling process and thus have an important influence on both net and gross NO₃⁻ uptake rates during ExLF.

However, NH₄⁺ concentrations sampled monthly remained consistently low (<0.1 mgN L$^{-1}$, less than one-tenth of the average NO₃⁻ concentration) and insufficient to provide additional information on N internal cycling. To advance the understanding of N processing during ExLF, higher-resolution NH₄⁺ concentration data would be beneficial.

Beyond assimilatory uptake, dissimilatory processes such as denitrification is also an important pathway for net NO₃⁻ uptake. In the Lower Bode, denitrification accounted for approximately one-third of net N uptake during the summer months (Huang

et al., 2022). Since $U_{NET-NO_3^-}$ did not change significantly between LF and ExLF, it follows that areal denitrification rates also remained relatively stable. During ExLF, the decline in NO₃⁻ concentrations could have reduced denitrification rates, whereas increased temperatures might have enhanced them (Seitzinger et al., 2006). These opposing effects may have offset each other, resulting in no significant change in the areal denitrification rates.

Despite the non-significant change in $U_{net}$–NO₃⁻, the overall NO₃⁻ input to the Lower Bode decreased significantly due to

lower NO₃⁻ concentrations and reduced flow during ExLF. Consequently, the percentage of NO₃⁻ removal within the study reach increased significantly, indicating improved nitrate removal efficiency. This enhanced NO₃⁻ removal efficiency also explains the greater spatial variability in NO₃⁻ concentrations between upstream and downstream observed in the Lower Bode (Section 3.4), suggesting increased spatiotemporal heterogeneity in solute concentrations (Hensley et al., 2019).



### 3.7 Implications of extreme summer low flow on water quality

The extreme summer low flow event of 2018 affected water quality in multiple aspects in the Lower Bode (Fig. 5). With climate change increasing the frequency and severity of droughts, extreme low flow conditions may become more common and intense in the future. More severe ExLF events would result in even lower water volumes, longer residence times, and higher solar radiation exposure, leading to prolonged periods of elevated water temperatures. One of the most immediate consequences would be an extended duration above the 25°C ecological threshold, posing a severe threat to the survival of

many aquatic organisms. Rising water temperatures could also trigger cascading effects on chemical reaction rates, gas solubility, biological productivity, and overall aquatic ecosystem health. For instance, DO levels could drop further—potentially falling below critical ecological thresholds—due to lower oxygen saturation and increased ecosystem respiration (ER).

Additionally, prolonged increases in water temperature could bring a growing threat of abrupt and irreversible shifts in

freshwater ecosystem functioning, potentially pushing systems beyond the tipping point. Flow-induced shifts in a stream community have been observed not only in whole ecosystem-scale experiments (Rosero-López et al., 2022) but also in European streams/rivers experiencing frequent summer extreme low flows in recent years. Notable examples include recurring cyanobacteria blooms in the Moselle River (Germany) since 2015 and massive proliferation of the brackish water algae *Prymnesium parvum* in the Oder River (Poland and Germany) in 2022. In contrast, during the ExLF event of 2018 in

the Lower Bode, while phytoplankton levels increased slightly, no bloom and ecosystem regime shift occurred. Instead, benthic algae, which have higher light availability and temperature tolerance, were the primary beneficiaries, leading to increased algal productivity. The modest rise in phytoplankton biomass during the 2018 ExLF suggests that the system had not yet reached a critical threshold for abrupt changes. However, if future ExLF events become more severe, they could push the system past a tipping point, triggering cascading ecological consequences such as ecosystem collapse, fish mortality,

water supply disruptions, and declines in recreational water quality.

Beyond direct temperature effects, reduced nutrient input from the catchment, coupled with enhanced gross primary production (GPP), could further accelerate internal nutrient processing. Such shifts may alter the role of streams and rivers in carbon, nitrogen, and phosphorus cycling, as well as impact food-web dynamics (Sabo et al., 2017).

Moreover, the compounding effects of extreme summer low flow events—such as drought-rewetting cycles and defoliation

of riparian vegetation—could amplify and prolong these impacts (Addy et al., 2018; Peña-Guerrero et al., 2020). Notably, the Lower Bode experienced ExLF conditions for three consecutive summers from 2018 to 2020. This study represents an initial step in understanding the effects of ExLF on water quality. Future research leveraging high-frequency sensor data and long-term monitoring will be crucial in determining whether cumulative ExLF events exert compounding effects on ecosystem function or if a tipping point has already emerged.

Hydrological and environmental changes during ExLF interact with the specific characteristics of stream and river systems, producing varied responses across different ecosystems. These complex interactions can either dampen or amplify the





sensitivity of in-stream processes and water quality responses to ExLF. Consequently, the extent and nature of ExLF-induced water quality changes are system-dependent, influenced by factors such as stream order, depth, morphology, canopy cover, aquatic ecosystem composition, and human activities (e.g., point-source pollution and flow regulation) (Hosen et al., 2019;

Mosley, 2015). Expanding research across diverse stream and river systems with different physical, biological, and anthropogenic influences will provide a more comprehensive understanding of the ongoing and future challenges facing global riverine ecosystems.

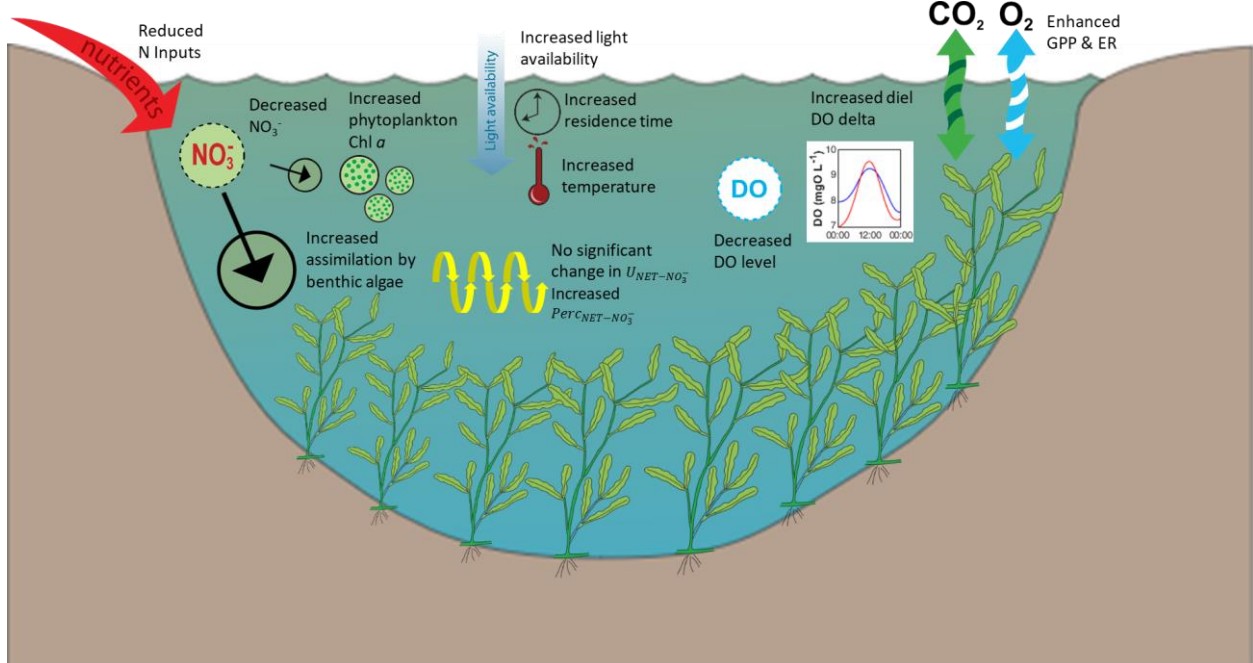

**Figure 5: Conceptual diagram summarising the changes in water quality variables and ecosystem processes during ExLF in the**
**Lower Bode**

### 4. Conclusion

Using multi-year high-frequency water quality measurements, we gained a comprehensive understanding of the water quality dynamics and ecosystem processes in the Lower Bode, a typical agricultural lowland stream in Central Germany, during the extreme low flow of summer 2018 (Fig. 5). Compared to summer low flows in 2014-2017, the 2018 ExLF

resulted in significantly warmer water temperatures, leading to an overall decrease in DO concentrations. However, the diurnal variation in DO levels increased substantially, indicating shifts in metabolic activity within the system. Both gross primary production (GPP) and ecosystem respiration (ER) were significantly enhanced during the ExLF, though the increase in GPP exceeded that of ER, making the system less heterotrophic. Elevated temperatures and increased light availability likely contributed to the boost in GPP. While phytoplankton Chl $a$ concentrations increased significantly, overall levels

remained low, suggesting that benthic algae were the primary drivers of enhanced ecosystem metabolism. $NO_3^-$ concentrations declined due to reduced catchment connectivity and improved in-stream $NO_3^-$ processing efficiency. Although the areal rate of net $NO_3^-$ removal did not change significantly, the percentage of $NO_3^-$ removed increased notably. This study provides robust *in situ* evidence—derived from high-frequency sensor data—of measurable shifts in water quality and ecosystem processes under extreme low flow conditions. Our findings underscore the importance of further research into

water quality responses to climate change, with broad implications for similar aquatic systems worldwide.

**Data availability**

The high-frequency water quality from the sensors used in this study are archived in the TERENO (Terrestrial Environmental Observatories) database and available for the scientific community upon request through the TERENO Online Data Portal (https://www.tereno.net/ddp; UFZ, 2025). The discharge data are freely available and

downloadable from the data portal of the State Office of Flood Protection and Water Quality of Saxony-Anhalt (https://gld.lhw-sachsen-anhalt.de/; LHW, 2025). The water quality data of the tributaries are also downloadable from the same portal. The hourly solar radiation data can be downloaded from open data portal of German Meteorological Service (https://opendata.dwd.de/; DWD, 2025).

**Author contribution**

JH conceptualized the paper, developed the methodology, validated the research, conducted the formal analysis and investigation, prepared the original draft preparation, and visualized the paper. DB collected the resources, reviewed and edited the paper, and acquired the funding. MR curated the data, reviewed and edited the paper, and supervised and administered the project.

**Competing interests**

The authors declare that they have no conflict of interest.

**Acknowledgements**

We thank the flood protection and water management agency of the state of Saxony-Anhalt, Germany (LHW), for providing data on water discharge and routine water quality data. We also thank the TERENO (Terrestrial Environmental Observatories) project for supporting the high-frequency monitoring in Lower Bode and Uwe Kiwel to maintain the sensor

measurements.



**Financial support**

Jingshui Huang was supported by the CSC-DAAD Postdoc Fellowship Program and Technical University of Munich (TUM) University Foundation Fellowship.

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
