# Peer review of "Changes in water quality and ecosystem processes at extreme summer low flow of 2018 with high-frequency sensors"

_EGUsphere, 2025_

## Author Response (AR1)

**Author's response**

**egusphere-2025-656**

Dear Reviewers,

Thank you very much for your valuable comments. The point-by-point responses to your comments are listed below.

**RC1**

This paper provides an evaluation of the effects of droughts on water quality with a special focus on oxygen and nitrogen processing. This contribution can be valuable to the field, after the authors carry out a major revision. Please follow my suggestions and comments below:

The title seems incomplete, e.g., "detected with high-frequency sensors"?

**Response:** Thank you for the comment. We have modified the title accordingly.

• Higher temperature will enhance stream metabolism and nutrient uptake but also dissolution of compounds in stream water – have you corrected these metrics for temperature effects before comparing them with 2014-2017 values? Specifically, have you corrected DO concentrations for temperature? Have you used flow-weighted concentrations for comparison? Please clarify.

**Response:** Thank you for this valuable comment. This is a very good observation. Indeed, we have considered the temperature effect on DO concentrations for comparison. As you can see in Table 1 in the manuscript, we have introduced daily DO saturation (DOS) and daily DO deficit (DOD) to consider the effect of temperature on DO dissolution. Stream metabolism calculations incorporated temperature-corrected saturated DO concentrations. However, we did not use flow-weighted concentrations for comparison.

Line 33 grammar

**Response:** Thank you for the comment. We have changed the sentence to "Drought events are a key driver to low-flow conditions in rivers and streams (Van Loon, 2015)."

• Sentence in lines 39-40 – logic is missing, "While the impacts of extreme low flows on water quantity are well-documented, there are still knowledge gaps regarding the effects on water quality..." – so are they well documented or are there still knowledge gaps?

**Response:** Perhaps there is a misunderstanding. The impact on water **quantity** is well-documented; however, the impacts on water **quality** are still facing knowledge gaps.

• Lines 50-60, please support with appropriate references

**Response:** Thank you for your suggestion. Two more references are added to the revised version now.

Graham, D. J., Bierkens, M. F. P., and van Vliet, M. T. H.: Impacts of droughts and heatwaves on river water quality worldwide, Journal of Hydrology, 629, 130590, https://doi.org/10.1016/j.jhydrol.2023.130590, 2024.

Li, L., Knapp, J. L. A., Lintern, A., Ng, G. H. C., Perdrial, J., Sullivan, P. L., and Zhi, W.: River water quality shaped by land—river connectivity in a changing climate, Nature Climate Change, 14, 225-237, 10.1038/s41558-023-01923-x, 2024.

Line 58 wording error

**Response:** Thanks for the comment. We have revised the sentence to "This fine-scale temporal resolution is crucial for understanding how ecosystems respond to transient events, such as extreme **high** flows, **during which** rapid changes can significantly influence water quality and ecosystem processes."

• Line 65 your list of publications covering the topic seems incomplete, please identify other publications. In general, there have been more publications coming out on these topics in the last years, please update your references as they seem a bit outdated.

**Response:** Thank you for your suggestion. We have checked more recent publications on the topics and have updated them in the revised version. Three recent publications are added as below:

Johnston, S. G., and Maher, D. T.: Drought, megafires and flood - climate extreme impacts on catchment-scale river water quality on Australia's east coast, Water Research, 218, 118510, https://doi.org/10.1016/j.watres.2022.118510, 2022.

Li, L., Knapp, J. L. A., Lintern, A., Ng, G. H. C., Perdrial, J., Sullivan, P. L., and Zhi, W.: River water quality shaped by land—river connectivity in a changing climate, Nature Climate Change, 14, 225-237, 10.1038/s41558-023-01923-x, 2024.

van Vliet, M. T. H., Thorslund, J., Strokal, M., Hofstra, N., Flörke, M., Ehalt Macedo, H., Nkwasa, A., Tang, T., Kaushal, S. S., Kumar, R., van Griensven, A., Bouwman, L., and Mosley, L. M.: Global river water quality under climate change and hydroclimatic extremes, Nature Reviews Earth & Environment, 4, 687-702, 10.1038/s43017-023-00472-3, 2023.

• Line 269 logic again — "As a lowland agricultural stream, the Lower Bode is not heavily influenced by point sources" — one can expect a strong impact of point sources in an agricultural setting, please clarify your reasoning here

**Response:** We appreciate this insightful comment and agree that agricultural streams can often be affected by point-source pollution. However, in the Lower Bode, our findings indicate that non-point sources (particularly agricultural runoff) dominate nutrient inputs, as supported by long-term water quality monitoring and modeling studies (e.g., Yang et al., 2018).

Key evidence for minimal point-source influence:

- 1. Low BOD inputs: Water quality data show consistently low biochemical oxygen demand (BOD) levels, inconsistent with significant point-source discharges.
- 2. DO patterns under low flow: If point sources were dominant, we would expect sustained oxygen depletion. Instead, we observe:
  - Pronounced diel DO cycles (elevated daytime peaks from photosynthesis, nighttime declines from respiration)
  - No systematic oxygen depression characteristic of point-source pollution (e.g., downstream of wastewater inputs)

To avoid confusion, we have also revised the text to: "As a lowland agricultural stream with minimal point-source inputs, the Lower Bode's DO dynamics are primarily governed by ecosystem processes (photosynthesis/respiration)."

• And the following sentence "Instead, its DO balance is governed by ecosystem processes such as photosynthesis and respiration." – do you have any evidence to support these two claims? How about groundwater influxes, have you accounted for them in your study?

**Response:** We thank the reviewer for these insightful questions regarding DO dynamics in our study system. Our conclusion about the dominance of ecosystem processes is supported by multiple lines of evidence:

- 1. Diurnal DO variability: High-resolution sensor data (see Fig.) show clear diurnal fluctuations—a hallmark of biologically driven DO cycling (e.g., peak DO during daylight hours due to photosynthesis, and nighttime declines from respiration).
- 2. We have done a parameter sensitivity of dissolved oxygen (DO) dynamics during the low-flow (LF) period in 2016 using the same model setup as this study using Elementary Effect (EE) method. This analysis revealed that benthic algae-related parameters (e.g., growth and respiration rates) exhibited the highest sensitivity, indicating their dominant influence on DO processes (Huang et al., 2021). These findings were initially presented in our EGU General Assembly poster contribution in 2021. A detailed ranking of all parameters, along with their definitions and units, is provided in the figure and table below.

Fig. 1 Parameter sensitivity ranking for DO concentration under low flow

**Table 1** Top 6 sensitive parameters for DO concentration under low flow

| Para.               | Definition                                | Units                              | Range      |
|---------------------|-------------------------------------------|------------------------------------|------------|
| F Gb20   | Benthic algae maximum growth rate         | gD m -2 d -1 | 5 – 100    |
| $K_Lb$              | Light constant for benthic algal growth   | Ly d -1                 | 50-300     |
| $\mathbf{k}_{Db20}$ | Benthic algae death rate                  | d -1                    | 0.001-0.2  |
| k Rb20   | Benthic algae respiration rate            | d -1                    | 0.05 - 0.2 |
| $\mathbf{k}_{Gmax}$ | Phytoplankton maximum growth rate at 20 C | d -1                    | 0.5 - 4.0  |
| k
deox           | Carbonaceous deoxygenation rate at 20 C   | d -1                    | 0.05 - 0.4 |

Regarding groundwater influences, we specifically analyzed water balance during the extreme summer low-flow period of 2018. Our calculations revealed a minimal water balance imbalance of just +0.59% (the positive value means that discharge at the outlet was lower than the input). This provides direct evidence that the direct exchange with groundwater of the main stem of the study reach in the Lower Bode is therefore very limited.

• I am not a big fan of mixing results with their discussion, please separate these to streamline the manuscript in a better way. At the moment, it is quite difficult to follow. Perhaps, more meaningful and less cheesy headings would be more suited. Please avoid using comparisons like "slight" or "slightly" – they dilute your message.

**Response:** We sincerely appreciate the reviewer's valuable feedback regarding manuscript organization. While we acknowledge that the traditional separation of Results and Discussion sections often enhances readability, we intentionally adopted an integrated approach for this study due to the following considerations:

- Interpretational Complexity: Our findings require immediate contextualization with process explanations and direct comparisons to existing literature to prevent potential misinterpretation of non-intuitive patterns. Maintaining these elements together reduces the need for excessive cross-referencing between sections and helps preserve the logical flow of our arguments.
- 2. Synthesis Section: We have included Section 3.7 specifically to integrate all water quality parameters and ecosystem processes into a comprehensive discussion, providing readers with a holistic understanding of our findings.

To address the reviewer's valid concerns, we have revised the subsection headings in Section 3 in the revised version. We have added a brief rationale in the last paragraph of the Introduction justifying the integrated structure in the revised version.

• Finally, what was truly novel about your approach? You simply repeat the same approach as in your 2016 paper. I am not convinced that extending your analysis to an extreme drought of 2018 is enough of a novelty.

**Response:** We appreciate the reviewer's attention to the methodological connections between this study and our earlier work (Huang et al., 2022). While both studies leverage some overlapping datasets and modeling foundations, the scope, focus, and advancements of this work are distinctly different:

- Expanded Scope: Unlike our 2022 study, which focused solely on nitrate processes, this
  work systematically examines multiple water quality parameters (e.g., temperature,
  dissolved oxygen, Chl-a) and ecosystem processes during an unprecedented extreme
  event—the 2018 drought. This broader perspective reveals interactions and responses that
  were not previously investigated.
- 2. Different Focus: In our 2022 study, we were interested in the seasonal variations of the nitrogen processes. As we were interested in the average changes, we did not separate 2018 extreme low flow from the results and analysis. In this study, we only focus on the summer low flows and extremes.
- 3. New Analysis: The data comparison using Kruskal–Wallis test including all water quality, environmental conditions, ecosystem processes were conducted newly only in this study.
- 4. Unique Findings: The water quality responses observed in 2018 could not have been extrapolated from our earlier work. Without the targeted reanalysis conducted here, these insights would remain inaccessible.

We acknowledge that some methodological continuity exists, but the combination of expanded parameters and tailored analysis provides novel contributions to understanding drought impacts on water quality.

The manuscript explores differences in water quality and metabolism dynamics in the Lower Bode between an extreme low-flow event and "normal" low-flow events. Metabolism is calculated using high-frequency in-situ data using the diel oxygen model. Calculated total GPP and inferred phytoplankton GPP rates are used to get benthic GPP. Nitrate uptake rates are quantified using a mass balance approach. The paper highlights the role of benthic algae in elevating GPP and gross nitrate uptake during extreme low-flow events. The paper summarizes its findings in a conceptual model how future, probably more common, extreme low-flow conditions will affect stream health. These results are of interest to freshwater ecologists and limnologists interested in future ecosystem changes.

**Main points:**

• Metabolism model: It would be great to show the diel model you have been using for metabolism calculations here. Also, you discuss later how temperature affects reaction rates using the Arrhenius equation, and you highlight a biological reaction rate increase of 1.7 – 3.3 % using a back-of-the-envelope approach. I wonder how your results would like if you'd account for temperature kinetics in eq. 1; e.g., GPP\_P = G\_P \* C\_PHY \* ROC \* z \* Theta^(T2-T1). This would result in lower benthic GPP and maybe more pronounced differences between low-flow and extreme low-flow years.

**Response:** We appreciate the reviewer's attention to the temperature effects on GPP by phytoplankton in our metabolism model. To clarify, our formulation already explicitly accounts for temperature dependence through the net growth rate term  $G_P$  for the phytoplankton, which is dynamically calculated as:

$$G_P = k_G - k_R - k_D$$

where for the phytoplankton growth rate kG incorporates:

- $k_G = k_{Gmax} \times X_T \times X_L \times X_N$
  - o kGmax: Maximum growth rate at 20°C, day-1 (0.5-4.0)
  - $_{\circ}$  XT: Temperature multiplier. We used Additional Temperature Function for phytoplankton growth calculation as:

$$\begin{split} X_T &= \exp[-\kappa_1(T_{opt}\text{-}T)^2] \quad \text{if } T \leq T_{opt} \\ &= \exp[-\kappa_2(T\text{-}T_{opt}\text{-}T)^2] \quad \text{if } T > T_{opt} \end{split}$$

- Topt: optimum temperature for growth, °C (10 27)
- $\kappa_1$ ,  $\kappa_2$ : temperature coefficients below and above optimum,  $1/^{\circ}C^2$  (0.005 0.04)
- o XL, XN: Light and nutrient limitation (range: 0-1)

In this case, the  $T_{opt}$ ,  $\kappa_1$ ,  $\kappa_2$  are assigned with the values of 13 degree Celsius, 0.02  $C^{-2}$  and 0.02  $C^{-2}$  respectively. Key points regarding temperature effects on phytoplankton growth are as follows:

- 1. The temperature response is fundamentally nonlinear, with distinct coefficients below/above  $T_{\text{opt}}$
- 2. All the above parameters were calibrated against observed chlorophyll-a dynamics (Huang et al., 2022), successfully reproducing seasonal patterns at STF stations and diurnal variability in phytoplankton activity.
- 3. This formulation provides more biological realism than a simple Arrhenius ( $\Theta$ ) correction because: It captures growth inhibition at supraoptimal temperatures and reflects species-specific thermal optima.

The GPP by benthic algae derives from: GPP\_benthic = GPP\_total - GPP\_phytoplankton. In this case, the temperature effect with temperature effects implicitly incorporated.

We have supplemented the details on how the G\_P is calculated and the temperature effect on it in Section 2.4 in the revised version and supplementary materials.

**Minor points:**

• L45: I don't think "increased solar radiation" can count as an environmental condition here, wouldn't the causal connection be reduced cloud cover (as environmental condition) causing less reflection of incoming solar radiation?

**Response:** Thank you for the comment. In the terminology of water quality modeling, for example in WASP, solar radiation is seen as an environmental condition. Therefore, we followed the common protocol. The solar radiation we mean here is the net solar energy reaching the water surface after accounting for atmospheric absorption and cloud effects. While we agree that reduced cloud cover is indeed the primary meteorological driver of increased solar radiation, from a water quality modeling perspective, it is the resultant radiation flux at the water surface that serves as the direct environmental condition affecting aquatic processes. To avoid confusion, we have changed "increased solar radiation" to "increased near-surface solar radiation".

• Fig2: So, the red lines in (c) represent then the low-flow conditions right, which are compared to the red lines during 2018 which were the extreme low-flow event? Could you please maybe color them differently and add a legend for clarification? Same for Figure 3

**Response:** Ok. We have revised the figures and their captions as the reviewer suggested.

L160: "Key metrics such as [...] were analysed using MATLAB."

**Response:** Thank you for the comment. We have revised the sentence accordingly.

• L177: Please state C\_PHY for consistency here as g C/m3 (which of course wouldn't affect any results)

**Response:** Thank you for the comment. Yes, we have stated  $C_PHY$  for consistency here as g  $C/m^3$  in the revised version.

Eq 2: Is travel time dynamic or constant in your model?

**Response:** Yes, the travel time in our model is dynamic. It is calculated with the hydrodynamic module in WASP.

Eq 3: Is I then the input loadings at t, hence all of US + TR?

Response: Yes, it is.

• L210: It's a bit confusing that delta isn't explained here but only visually in Fig. 4. I first thought that you mean the difference between two measurements here. Is the O2 deficit between 100% saturation and measured saturation conc.? What is Chl-a accumulation?

**Response:** We thank the reviewer for highlighting this ambiguity. We recognize these terms require clearer definition. Daily DO delta (DO $\Delta$ ) is calculated as the difference between the daily DO maximum and minimum. The DO deficit is defined as the difference between 100% DO saturation concentration and measured DO concentration. The Chl-a accumulation is defined as the difference between the downstream Chl-a concentration and the upstream concentration, reflecting net algal growth over the study reach. These descriptions on certain terms have been supplemented to section 2.3 and the caption of Table 1.

• L225: What do you mean with "thermal capacity", capacity related to biota like in Lake 2003 or the specific heat capacity of water (which wouldn't be affected), or heat storage in reduced volumes?

**Response:** We thank the reviewer for this important clarification. Here, "thermal capacity" refers specifically to the heat storage potential of the reduced water volume, not the specific heat capacity of water (which remains constant) nor biotic tolerance thresholds. We recognize this could be misinterpreted and have revised the text to "thermal buffering capacity". In addition, the reference by Lake (2003) is removed to avoid conceptual conflation.

• Table 1: Are the values given for LF and ExLF the averages across the individual seasons? Hence, is ExLF of 2018 compared to the average behavior of all LF's before? If indeed these are averages, maybe also give standard deviations or quantiles to make full use of your high-frequency data. It's a bit contradicting that you praise high-frequency data for metabolism calculations but then show only a value of each season/event plus the statistical p-value. Also, I think you should expand the caption of the table to explain all variables again as otherwise the reader has to go back to the text every time.

**Response:** Thank you for your valuable suggestions. Yes, the values in the table are the medians of 2018 seasons for ExLF and medians of the 2014-2017 seasons. We have mentioned this explicitly in the table caption. The reviewer mentioned standard deviations or quantiles of the datasets. This is sensible. We have included them in Table S1 in the revised version. Besides,

we have expanded the captions of Table 1 and Table S1 to explain all variables in it to increase readability in the revised manuscript.

• L312: "increased", do you mean "suggested that increased phytoplankton growth rates led to higher [...]"? But isn't that related as higher growth rates cause higher conc.?

**Response**: Thank you for the reviewer to mention this. Yes, the sentence should be "suggested that increased phytoplankton growth rates led to higher..."

• L347: Wouldn't areal ER always indicate oxygen consumption and should be always negative?

**Response:** Yes, you are absolutely correct that ER indicates oxygen consumption and should be always negative. We have explicitly clarified this in the Methods section 2.4 about our sign convention: "Following standard aquatic metabolism conventions, positive GPP values indicate oxygen production while negative ER values represent oxygen consumption."

• L350: But can your analysis for sure determine if conditions were "less heterotrophic" as the p-value is not significant?

**Response:** Thank you for the comment and observation. Yes, indeed the NEP values between ExLF and LF were not significantly different according to the p-value. The saying of "less heterotrophic" came from the comparison of the median values of the NEP. We have modified the text as follows: "Though not significant (p = 0.45), the observed median NEP values suggested a potential trend toward reduced heterotrophy during ExLF conditions (Table 1)."

• L353 and onwards: I am missing a discussion of atmospheric exchange rates here in affecting overall NEP rates. Are they negligible and indeed all O2 changes can be attributed to GPP and ER?

**Response:** We appreciate the reviewer's important question regarding the role of atmospheric exchange in our NEP calculations. We would like to clarify and expand on how reaeration was incorporated into our analysis and its potential impacts during ExLF.

In our methodology (Section 2.4), we explicitly accounted for atmospheric exchange by estimating the reaeration rate using the O'Connor-Dobbins formula, which is particularly suitable for slow-flowing streams (Chapra, 2008). This approach incorporates key hydraulic variables including water depth and flow velocity, obtained from both gauging station measurements and hydrodynamic modeling results (Huang et al., 2022). Additionally, dissolved oxygen saturation levels were carefully determined based on water temperature, salinity, and barometric pressure measurements following standard methods (APHA, 1998).

During ExLF conditions, we recognize that atmospheric exchange becomes particularly influential on NEP calculations due to two primary factors: First, the reduced flow velocity decreases turbulence, leading to lower reaeration coefficients (k2). This results in slower oxygen exchange with the atmosphere, making dissolved oxygen dynamics more sensitive to

biological processes (GPP and ER). Second, the diminished gas exchange can amplify observed DO swings - potentially exaggerating daytime peaks from GPP and nighttime declines from ER.

While our method does account for changes in velocity and depth during ExLF when estimating reaeration rates, we acknowledge that NEP calculations become more sensitive to reaeration estimates under these extreme conditions. Any inaccuracies in reaeration estimation could introduce bias in our NEP results, potentially explaining why some of our NEP comparisons between ExLF and LF conditions showed non-significant differences. For instance, an overestimated reaeration rate could make the system appear more autotrophic, while an underestimation could bias results toward more heterotrophy.

The above-mentioned discussion is now supplemented in Section 3.5 in the revised manuscript, also shown below.

During ExLF conditions, atmospheric exchange plays an important role in shaping diel oxygen dynamics. For instance, overestimated reaeration could mask true heterotrophic conditions by attributing more DO gain to atmospheric input rather than biological production. Conversely, underestimation could exaggerate DO losses and skew NEP toward respiration. While we accounted for hydraulic changes in our reaeration rate estimation, we acknowledge that NEP during ExLF may be more sensitive to uncertainties in reaeration than under typical low-flow conditions. These dynamics may help explain why NEP differences between LF and ExLF were statistically non-significant, despite clear trends in GPP and ER. Future work may benefit from direct measurements of reaeration or sensitivity analyses to further constrain its influence under low-flow extremes.

• L356: Phytoplankton growth and GPP\_P are inherently linked in eq. 1, so isn't this expected? Would growth rates benefit only because biomass is elevated due to lower flushing rates?

Response: We appreciate the reviewer's observation regarding the relationship between phytoplankton growth and GPP\_P in our model. The reviewer is correct that phytoplankton growth rate (units of d-1) and GPP\_P are inherently linked through Eq. 1. However, we would like to clarify that while the growth rate itself was not directly enhanced by lower flushing rates, the longer residence time during extreme low-flow (ExLF) conditions did lead to greater phytoplankton biomass accumulation along the study reach. This increased biomass (C\_PHY in Eq. 1) subsequently resulted in higher GPP\_P values during ExLF periods.

To improve accuracy, we have revised the term "phytoplankton growth" to "phytoplankton biomass accumulation" throughout the manuscript to better reflect this distinction in the revised version.

L364: Could you add GPP B to Table 1 please.

**Response:** Yes, Ok. We have included it in Table 1 in the revised version.

L433: Would solar exposure be really higher or just longer?

**Response:** We thank the reviewer for raising this important question regarding solar exposure during ExLF. In this study, we analyzed long-term observational data of hourly shortwave global radiation (J/cm²) obtained from the German Weather Service. These radiation measurements were converted to Langley per day (Ly/d) to enable standardized comparison between LF and ExFL, as shown in Table 1.

Our results demonstrate that the total solar energy, represented by the hourly sum of shortwave global radiation, was significantly higher during ExLF conditions compared to LF periods. This metric reflects the integrated product of both solar irradiance intensity and duration. Here, solar exposure means total solar energy received over time, which could be represented by this metric. So, we can draw conclusions on the solar exposure are really higher. However, we acknowledge that our current analysis does not determine whether this increased solar exposure resulted from higher irradiance intensity, longer sunshine duration, or some combination of both factors. But one could incorporate the duration comparison as the German Weather Service also provide this parameter.

• L447: Could nutrient limitation or photosensitivity also play a role here that could be discussed?

**Response:** We appreciate the reviewer's suggestion regarding additional factors that could influence our results. In response to the insightful comment:

Regarding nutrient limitation, our WASP model simulations do indeed provide information about nutrient constraints during both LF and ExLF periods.

As mentioned in Section 3.5, N did not appear to be a limiting factor for GPP in the Lower Bode, as GPP continued to increase despite lower N concentrations during ExLF.

According to the WASP model results, the nutrient limitation factor is 0.946 at LF and 0.943 at ExLF. The percentage change is 0.3% lower than LF. The influence is 0.3%, showing benthic algae is not very sensitive to the ambient nutrient concentration change. This is also because the benthic algae nutrient limitation is calculated with internal storage of nutrients. The internal storage buffers the change of the ambient nutrient concentration. Compared to the total percentage increase in GPP $_{\rm B}$  during ExLF, i.e., 53%, this decrease is minimal.

We have incorporated this analysis into Section 3.5 in the revised discussion to examine how nutrient availability may have interacted with the observed temperature and light effects, shown also as below.

This interpretation is supported by our previous WASP model simulations (Huang et al., 2022), which estimated a nutrient limitation factor of 0.946 during LF and 0.943 during ExLF—indicating only a 0.3% reduction (Fig. S4). This minimal change suggests that nutrient availability was not a primary constraint on benthic algal productivity. Furthermore, benthic algae in the Lower Bode are modeled with internal nutrient storage, which buffers short-term fluctuations in ambient concentrations (Droop, 1973). This internal regulation likely helped maintain high productivity despite reduced external N inputs. Thus, while nutrient availability

was slightly lower, its influence on GPPB was marginal compared to the strong positive effects of increased light penetration and thermal conditions.

Concerning photosensitivity, while our current modeling framework includes light limitation through the Half-Saturation equation, we acknowledge that alternative formulations (such as Smith or Steele equation) could potentially yield different results. However, as we did not conduct sensitivity analyses comparing these different light limitation approaches in this study, we are unable to rigorously evaluate the potential role of photosensitivity effects at this time.

**References**

Yang, X., Jomaa, S., Zink, M., Fleckenstein, J. H., Borchardt, D., & Rode, M.: A new fully distributed model of nitrate transport and removal at catchment scale. *Water Resources Research*, 54, 5856–5877, <a href="https://doi.org/10.1029/2017WR022380">https://doi.org/10.1029/2017WR022380</a>, 2018.

Huang, J., Merchan-Rivera, P., Chiogna, G., Disse, M., and Rode, M.: Can high-frequency data enable better parameterization of water quality models and disentangling of DO processes?, EGU General Assembly 2021, online, 19–30 Apr 2021, EGU21-8936, https://doi.org/10.5194/egusphere-egu21-8936, 2021.

Huang, J., Borchardt, D., and Rode, M.: How do inorganic nitrogen processing pathways change quantitatively at daily, seasonal, and multiannual scales in a large agricultural stream?, Hydrol. Earth Syst. Sci., 26, 5817–5833, https://doi.org/10.5194/hess-26-5817-2022, 2022.

---

## Referee Report (RR1)

Review of "Changes in water quality and ecosystem processes at extreme summer low flow of 2018 detected with high-frequency sensors" Huang et al. at HESS (ref: egusphere-2025-656)

The study demonstrates that the extreme low-flow conditions of summer 2018 in the Lower Bode stream led to marked alterations in some water quality and ecosystem functioning parameters. Elevated water temperature and chlorophyll-a concentrations coincided with reduced dissolved oxygen and nitrate levels. Stronger diurnal oxygen fluctuations and a significant increase in gross primary productivity, dominated by benthic algae, were observed alongside higher ecosystem respiration, resulting in near-zero net ecosystem productivity. Although less clear, net nitrate uptake rates did not change, the proportion of nitrate removed increased significantly due to benthic algae assimilation, indicating a more efficient internal nutrient cycle during extreme drought conditions.

The manuscript provides novel insights by employing high-frequency, reach-scale measurements to assess ecosystem responses under extreme low flow, a methodological approach still rare in the literature compared with studies based on traditional grb sampling schemes. This study adds to a growing body of recent research of drought effects on aquatic hydrology, ecology and biogeochemistry by providing novel insights into water quality and instream ecosystem processes under extreme low-flow conditions. It is both original and significant, as it enhances our understanding and predictive capacity regarding the consequences of more frequent and severe droughts in Central Europe under climate change, with clear implications for freshwater ecosystem management.

Overall the ms. is very clearly presented, well-structured and relies on highly valuable, high quality data.

**Major comments**

- The relative simplicity of the comparative analysis between drought and extreme summer drought conditions makes the results easy to follow and convincing. However, I believe that a Q-C and/or hysteresis-type analysis could help to better understand the sensitivity of each site, water quality parameter or ecosystem process to changing flow conditions, as well as the trajectory of these responses during flow reduction (in a drought) and flow recovery (after the drought).
- One of the paper's most innovative goals is to exploit cutting-edge sensor technologies to more effectively capture the rapid and novel mechanisms underlying water quality and ecosystem functioning responses under low-flow conditions. However, one of the major challenges is to properly calibrate these sensors. While this issue has already been resolved for some parameters included in the study, for others it remains quite complex and requires a solid set of 'classical measurements' taken in the field and covering environmental gradient comparable to those of the study. Although the paper does mention this aspect, it lacks a detailed description of the protocols followed to calibrate the *Chl-a* and NO3- sensors. For these variables, I also find the absence of a 1:1 plot comparing sensor-based measurements with classical sampling and laboratory analyses.
- The manuscript provides a description of in-stream aerobic metabolism modeling, but the presentation lacks sufficient detail on key aspects of the model and the

results obtained. Uncertainties in the estimates are mentioned, yet the sources of variability and how they influence the results are not fully explored. While the potential integration of lateral oxygen inflows is briefly discussed, the evaluation remains superficial and does not convincingly demonstrate their impact. Alongside the previous, some examples of observed versus modeled dissolved oxygen concentrations should be included in the supplementary information.

**Minor comments**

Lines 96-101\*: This level of detail, including the description of the statistical tests used, is not meant to be included in the introduction.

Line 243: Panel letters of Figure 3 are missing in the Figure but referenced in the text.

Line 293: Correct: "at GGL by 0.45 mg L-1 at GGL (p < 0.01) and non-significantly at STF 0.28 mg L-1":

Line 295: add by between "and" and "0.73".

Line 370: remove mobile

Line 377: expand the how in-stream processes can affect/are affecting NO3- removal. What about other dissolved inorganic N forms.

The following key references on this topic are missing (Gómez-Gener et al. 2020; Dupas et al. 2025; Harjung):

Dupas, R., A. Lintern, A. Musolff, C. Winter, O. Fovet, and P. Durand. 2025. Water quality responses to hydrological droughts can be predicted from long-term concentration—discharge relationships. Environ. Res.: Water 1: 015001. doi:10.1088/3033-4942/adb906

Gómez-Gener, L., A. Lupon, H. Laudon, and R. A. Sponseller. 2020. Drought alters the biogeochemistry of boreal stream networks. Nat Commun **11**: 1795. doi:10.1038/s41467-020-15496-2

Harjung, A. Impact of drought periods on carbon processing across surface-hyporheic interfaces in fluvial systems. 232.

\* Line numbers corresponds to author track changed document.

---

## Author Response (AR2)

**Author's response**

**egusphere-2025-656**

Dear Editor and Reviewers,

Thank you very much for your valuable comments. The point-by-point responses to your comments are listed below.

**Report #1**

The authors did incorporate and build upon my initial comments and thoughts. I especially enjoy reading the new discussions on atmospheric exchange and nutrient limitation. Also, the additional information in the supplement is now sufficient to understand the modeling.

Still, I would heavily suggest that the authors revise Fig. 2 and 3 by adding a legend that explains what the blue and red lines represent (this would make it easier for a reader who skipped the text). Only Fig. 2 explains the red line.

**Response:** Thank you for the acknowledgement of our responses to the review in the last round. Regarding the suggestion of Fig. 2 and 3, we now added a legend that explains what blue and red lines represent. Thanks for the suggestion.

**Report #2**

The study demonstrates that the extreme low-flow conditions of summer 2018 in the Lower Bode stream led to marked alterations in some water quality and ecosystem functioning parameters. Elevated water temperature and chlorophyll-a concentrations coincided with reduced dissolved oxygen and nitrate levels. Stronger diurnal oxygen fluctuations and a significant increase in gross primary productivity, dominated by benthic algae, were observed alongside higher ecosystem respiration, resulting in near-zero net ecosystem productivity. Although less clear, net nitrate uptake rates did not change, the proportion of nitrate removed increased significantly due to benthic algae assimilation, indicating a more efficient internal nutrient cycle during extreme drought conditions.

The manuscript provides novel insights by employing high-frequency, reach-scale measurements to assess ecosystem responses under extreme low flow, a methodological approach still rare in the literature compared with studies based on traditional grb sampling schemes. This study adds to a growing body of recent research of drought effects on aquatic hydrology, ecology and biogeochemistry by providing novel insights into water quality and instream ecosystem processes under extreme low-flow conditions. It is both original and significant, as it enhances our understanding and predictive capacity regarding the

consequences of more frequent and severe droughts in Central Europe under climate change, with clear implications for freshwater ecosystem management.

Overall the ms. is very clearly presented, well-structured and relies on highly valuable, high quality data.

**Response:** Thank you for the nice summary of the manuscript.

**Major comments**

The relative simplicity of the comparative analysis between drought and extreme summer
drought conditions makes the results easy to follow and convincing. However, I believe that
a Q-C and/or hysteresis-type analysis could help to better understand the sensitivity of
each site, water quality parameter or ecosystem process to changing flow conditions, as
well as the trajectory of these responses during flow reduction (in a drought) and flow
recovery (after the drought).

**Response:** Thank you for the valuable comment. As the reviewer noted, our comparative design was intended to provide clear and direct evidence of the impact of extreme summer low flow on water quality. We agree that C–Q and hysteresis analyses are powerful tools for quantifying site sensitivity and for examining the trajectories of water quality responses during drought onset and recovery. While our dataset would allow such analyses, this study focuses specifically on a direct comparative framework to highlight the effects of extreme low-flow conditions in a straightforward and convincing manner. A full C–Q or hysteresis-based investigation would require a different analytical design and is therefore beyond the scope of the present paper.

In recognition of the relevance of this suggestion, we have elaborated Section 3.7 to emphasize this as an important direction for future work as follows: Moreover, future research is encouraged to leverage high-frequency datasets and apply C–Q and hysteresis approaches to assess short-term nutrients and ecosystem dynamics and to investigate the underlying mechanisms during both drought onset and recovery, thereby extending the insights gained from our comparative analysis.

• One of the paper's most innovative goals is to exploit cutting-edge sensor technologies to more effectively capture the rapid and novel mechanisms underlying water quality and ecosystem functioning responses under low-flow conditions. However, one of the major challenges is to properly calibrate these sensors. While this issue has already been resolved for some parameters included in the study, for others it remains quite complex and requires a solid set of 'classical measurements' taken in the field and covering environmental gradient comparable to those of the study. Although the paper does mention this aspect, it lacks a detailed description of the protocols followed to calibrate the *Chl-a* and NO3- sensor-based measurements with classical sampling and laboratory analyses.

**Response:** Thank you for this helpful comment. Both sensors were routinely calibrated on a monthly basis as part of our maintenance program. For  $NO_3^-$ , we compared sensor readings with laboratory-analyzed grab samples and applied corrections following the established methodology of Rode et al. (2016) from the Bode observatory in the Selke River. In that study, the method achieved an  $R^2$  of 0.93 with low bias at Station Meisdorf. In our case, we also provide a comparison of sensor and laboratory measurements, together with the corrected  $NO_3^-$  data for the GGL station in the Supplement (Fig. S5), which demonstrates the good agreement between the original sensor measurements and the grab samples, with only minor bias.

Sensor data were corrected against grab samples using our automated MATLAB data-cleaning tool, which shifts the continuous sensor signal to align with laboratory results when necessary. This step is particularly important for  $NO_3^-$  because of the characteristics of the UV-Vis absorbance method: the optical path length is fixed during measurements, while the optimal path length depends on concentration (e.g., ~1 cm for low concentrations and ~0.2 cm for high concentrations). When concentrations vary strongly, measurement uncertainties may occur if the path length remains constant. Corrections against laboratory data are therefore especially relevant during rare periods of elevated  $NO_3^-$  concentrations. We apply this correction only to nitrate for the above methodological reason, and an example of this adjustment is shown in Fig. S5 for the GGL Station.

Fig. S5 Comparison of nitrate ( $NO_3^--N$ ) concentrations measured by sensor and laboratory analyses, including data adjustments at the GGL station.

For *Chl-a*, sensor measurements and laboratory grab samples collected at STF between 2011 and 2019 showed very good agreement, with an R2 of 0.85 (also provided in SI as Fig. S6).

**Fig. S6** Relationship between *Chl-a* concentrations measured by sensors and laboratory grab samples at the STF station from 2011 to 2019.

In the revised manuscript, additional text has been included in Section 2.3 to address this point.

The manuscript provides a description of in-stream aerobic metabolism modeling, but the presentation lacks sufficient detail on key aspects of the model and the results obtained. Uncertainties in the estimates are mentioned, yet the sources of variability and how they influence the results are not fully explored. While the potential integration of lateral oxygen inflows is briefly discussed, the evaluation remains superficial and does not convincingly demonstrate their impact. Alongside the previous, some examples of observed versus modeled dissolved oxygen concentrations should be included in the supplementary information.

**Response:** Thank you for the comment. In this study, we applied the single-station method, which relies on high-frequency dissolved oxygen (DO) time series. In this approach, DO is measured directly by sensors and not modeled as an output; instead, the measured DO serves as the basis for estimating ecosystem metabolic rates. Therefore, no direct comparison between modeled and observed DO concentrations is available within this study. However, in our previous work (Huang et al., 2022), DO was modeled using the WASP framework for the same reach and study period. Comparisons of modeled and observed DO concentrations for the full 5-year period and the ExLF phase can be found in Figures 2(e) and S5 of that publication.

**Minor comments**

• Lines 96-101\*: This level of detail, including the description of the statistical tests used, is not meant to be included in the introduction.

**Response:** Thank you for the comment. We have deleted the details of statistical tests used in the revised manuscript.

• Line 243: Panel letters of Figure 3 are missing in the Figure but referenced in the text.

**Response:** Thank you for the comment. The panel letters of Figure 3 are added in the revised manuscript.

 Line 293: Correct: "at GGL by 0.45 mg L-1 at GGL (p < 0.01) and non-significantly at STF 0.28 mg L-1":

**Response:** Thank you for the comment. We have corrected the text in the revised manuscript.

• Line 295: add by between "and" and "0.73".

**Response:** Thank you for the comment. We have added it to the revised manuscript.

• Line 370: remove mobile

**Response:** Thank you for the comment. We have removed it from the revised manuscript.

• Line 377: expand the how in-stream processes can affect/are affecting NO3- removal. What about other dissolved inorganic N forms.

**Response:** Thank you for the valuable comment. Now we have expanded the discussion of instream processes affecting nitrate removal in Section 3.4 as below.

Beyond reduced lateral NO3- loading, enhanced instream processing could have further amplified spatial heterogeneity between upstream and downstream. Nitrate dynamics in streams are governed by three main instream processes: assimilation, denitrification (the only permanent removal pathway), and regeneration through nitrification. Under ExLF conditions, water residence time in the reach increased by 45% (Table 1), and reaction rates were enhanced by 7.1% at STF due to higher temperatures (based on a simplified Arrhenius calculation, Section 3.1). Although mean NO₃⁻ concentrations decreased by 9.5% at STF (Table 1), these hydrological and thermal changes likely enhanced denitrification and overall NO₃⁻ removal. Furthermore, as discussed in Section 3.6, the significant 46% increase in areal GPP during ExLF suggests that gross NO₃⁻ uptake rates also increased. Nitrification is likewise expected to intensify under longer residence time and elevated temperature, potentially increasing NO3- production. Indeed, NH4+ concentrations were higher during ExLF than LF (Fig. S1 in Huang et al. (2022)), further supporting higher NO₃⁻ production via nitrification. However, our NH4+ dataset is limited to monthly sampling, which prevents a robust quantification of this process. Notably, elevated NH4+ concentrations have also been reported in northern boreal streams during the 2018 drought (Gómez-Gener et al., 2020). Among these pathways, gross NO₃⁻ uptake appears to be the dominant driver. Long-term model estimates for this reach indicate a mean gross NO3- uptake rate is 60.5 mg N m-2 d-1, compared with a mean denitrification rate of 14.1 mg N m-2 d-1 (Table S3 in Huang et al. (2022)). Thus, enhanced assimilatory uptake under ExLF likely played a decisive role in net NO3- removal. Although nitrification may also have increased, NH4+ concentrations remained below 0.1 mg N L-1—far lower than NO₃⁻ concentrations—suggesting that the magnitude of nitrate regenerated was smaller than the combined removal via uptake and denitrification. Overall, this balance indicates that the increases in sink processes exceeded source processes during ExLF, which likely contributed to the greater upstream—downstream disparity in  $NO_3^-$  concentrations. This pattern aligns with findings by Hensley et al. (2019) and is consistent with broader evidence for reductions in  $NO_3^-$  concentrations during droughts in streams and rivers through diminished catchment inputs and enhanced in-stream retention (Caruso, 2001; Dupas et al., 2025; Mosley et al., 2012; Muchmore and Dziegielewski, 1983).

• The following key references on this topic are missing (Gómez-Gener et al. 2020; Dupas et al. 2025; Harjung):

Dupas, R., A. Lintern, A. Musolff, C. Winter, O. Fovet, and P. Durand. 2025. Water quality responses to hydrological droughts can be predicted from long-term concentration—discharge relationships. Environ. Res.: Water 1: 015001. doi:10.1088/3033-4942/adb906

Gómez-Gener, L., A. Lupon, H. Laudon, and R. A. Sponseller. 2020. Drought alters the biogeochemistry of boreal stream networks. Nat Commun **11**: 1795. doi:10.1038/s41467-020-15496-2

Harjung, A. Impact of drought periods on carbon processing across surface-hyporheic interfaces in fluvial systems. 232.

**Response:** Thank you for mentioning the key references. They are very helpful. Now we have cited the first 2 references in the revised manuscript in Section 3.4 and 3.3.

**References**

Rode, M., Halbedel née Angelstein, S., Anis, M. R., Borchardt, D., and Weitere, M.: Continuous In-Stream Assimilatory Nitrate Uptake from High-Frequency Sensor Measurements, Environmental Science & Technology, 50, 5685-5694, 10.1021/acs.est.6b00943, 2016.

Huang, J., Borchardt, D., and Rode, M.: How do inorganic nitrogen processing pathways change quantitatively at daily, seasonal, and multiannual scales in a large agricultural stream?, Hydrol. Earth Syst. Sci., 26, 5817-5833, 10.5194/hess-26-5817-2022, 2022.